# Regulation of two motor patterns enables the gradual adjustment of locomotion strategy in *Caenorhabditis elegans*

Ingrid Hums, Julia Riedl, Fanny Mende, Saul Kato, Harris S Kaplan, Richard Latham, Michael Sonntag[†], Lisa Traunmüller[‡], Manuel Zimmer*

Research Institute of Molecular Pathology, Vienna Biocenter VBC, Vienna, Austria

**Abstract** In animal locomotion a tradeoff exists between stereotypy and flexibility: fast long-distance travelling (LDT) requires coherent regular motions, while local sampling and area-restricted search (ARS) rely on flexible movements. We report here on a posture control system in *C. elegans* that coordinates these needs. Using quantitative posture analysis we explain worm locomotion as a composite of two modes: regular undulations versus flexible turning. Graded reciprocal regulation of both modes allows animals to flexibly adapt their locomotion strategy under sensory stimulation along a spectrum ranging from LDT to ARS. Using genetics and functional imaging of neural activity we characterize the counteracting interneurons AVK and DVA that utilize FLP-1 and NLP-12 neuropeptides to control both motor modes. Gradual regulation of behaviors via this system is required for spatial navigation during chemotaxis. This work shows how a nervous system controls simple elementary features of posture to generate complex movements for goal-directed locomotion strategies.

*For correspondence: zimmer@imp.ac.at

Present address: [†]Department Biologie II, Ludwig-Maximilians-Universität München, Planegg-Martinsried, Germany; [‡]Biozentrum, University of Basel, Basel, Switzerland

Competing interests: The authors declare that no competing interests exist.

## Introduction

Animals execute neural control over their motor systems to generate a multitude of behaviors such as grooming, courtship or foraging. These behavioral strategies often require very distinct types of motor patterns while employing the same muscle groups. For example, different modes of locomotion serve the optimal strategy for food finding. Under the assumption that food is nearby, a large variety of invertebrate and vertebrate species including mammals perform local or area-restricted search (ARS) consisting of short moves and frequent high-angled turns. Alternatively, in order to localize more distant food sources, animals disperse via long-distance travelling (LDT) by moving along straight paths (*Bell, 1990*; *Fryxell et al., 2008*; *Hills, 2006*). Understanding how nervous systems operate these different strategies of foraging behavior requires a detailed description of the underlying motor patterns as well as mechanistic insights into the neural circuits coordinating them.

We addressed these problems through investigations of the nematode *C. elegans*, which is a tractable genetic model organism with an anatomically mapped small nervous system and which employs a variety of strategies to navigate through its environment. Nematodes advance via undulatory crawling movements (*Gray, 1953*), which can be precisely quantified (*Brown et al., 2012*; *Stephens et al., 2008*; *Yemini et al., 2013*). The animal's posture is tightly coupled to ventral and dorsal body wall muscle activity and therefore a reliable proxy to characterize the motor patterns underlying gait production (*Butler et al., 2014*). When removed from food *C. elegans* performs frequent reorientation maneuvers, which consist of brief periods of backward crawling (reversals) followed by sharp turning (omega turns); however, when no food is detected for a prolonged time *C. elegans* maintains forward crawling and only infrequently reorients. These two locomotion strategies have been explicitly characterized as ARS after removal from food and LDT (*Calhoun et al., 2015*;

**eLife digest** Animals navigate through their environment using different strategies according to their current needs. For example, when the goal is to travel long distances, they move quickly and in an efficient way by employing regular, repetitive movements. However, when the aim is to explore the nearby area – to search for food, for example – animals move slowly and make more flexible movements. These different types of movement mostly use the same groups of muscles, and so animals must be able to alter how they control their muscles to yield these different strategies.

These movement strategies have been observed in many animal species, from worms to grazing cows, and researchers have mostly classified them into distinct behavioral states that the animals switch between. To date, the patterns of movements that underlie these strategies have not been described in detail.

The wavelike movement of the roundworm *Caenorhabditis elegans* has the advantage of being relatively easy to measure. By analyzing precise recordings of how the worms change posture as they move, Hums et al. now show that two main patterns of motion underlie worm movement. Regular whole-body waves (undulations) efficiently drive long-distance travel, while more complex turning motions allow the animals to flexibly change direction and so explore the local environment. Furthermore, the worms can fine-tune their movement strategy by gradually transitioning between the two patterns. This finding is opposed to the standard view, where animals switch between distinct behavioral states.

Hums et al. then studied how neuronal regulation in the *C. elegans* nervous system enables the worms to transition between the different movement strategies. In these experiments, neurons were manipulated and their activity was recorded. The results suggest that two classes of so called interneurons enable the worms to fine-tune their movements. Each class of these interneurons produces a signaling molecule (or neuropeptide) that counteracts the activity of the other signal; together both neuropeptides regulate the patterns of movements.

Further work is now needed to identify and investigate the downstream neurons that work together to represent the different patterns of movements in the roundworm. Future studies could also analyze whether other animals – such as swimming animals and limbed animals – use similar principles to change between distinct forms of movement and thus enact a range of behavioral strategies.

*Gray et al., 2005*; *Hills et al., 2004*; *Peliti et al., 2013*; *Tsalik and Hobert, 2003*; *Wakabayashi et al., 2004*). Also, two major locomotion strategies for goal-directed chemotaxis have been observed: the biased random walk, which is reminiscent of bacterial chemotaxis (*Berg and Brown, 1972*), where animals modulate the probability of reorientation depending on sensory history (*Pierce-Shimomura et al., 1999*), and weathervaning, where animals directly steer toward the favored direction by introducing a subtle bias to their path (*Iino and Yoshida, 2009*). The above-mentioned studies have quantified locomotion by the frequency and duration of forward runs, reversals and omega turns; however, quantitative descriptions of the motor patterns underlying the different locomotion strategies of *C. elegans,* or those of any animal species in general, have been missing.

Oxygen ($O_2$) chemotactic responses of *C. elegans* are an effective experimental paradigm to investigate the neural control of locomotion. In nature, *C. elegans* is found in association with anthropogenic soil habitats constituting complex environments rich in olfactory cues (*Félix and Braendle, 2010*; *Félix and Duveau, 2012*), in which local $O_2$ concentrations vary and might serve to indicate the presence of bacterial food, pathogens or predators (*Sexstone et al., 1985*). In the laboratory, worms are fed on *E. coli* lawns, which generate steep $O_2$ gradients ranging from atmospheric 21% down to 12% $O_2$ (*Gray et al., 2004*). *C. elegans* possesses sophisticated sensory systems to detect $O_2$ (*Busch et al., 2012*; *Gray et al., 2004*; *Zimmer et al., 2009*). A sudden decline in $O_2$ levels activates BAG class $O_2$ sensory neurons (*Zimmer et al., 2009*), and during navigation of spatial $O_2$ gradients, animals accumulate at intermediate concentrations (*Cheung et al., 2005*; *Gray et al., 2004*; *Zimmer et al., 2009*). Interneuron circuits that are involved in locomotory responses to

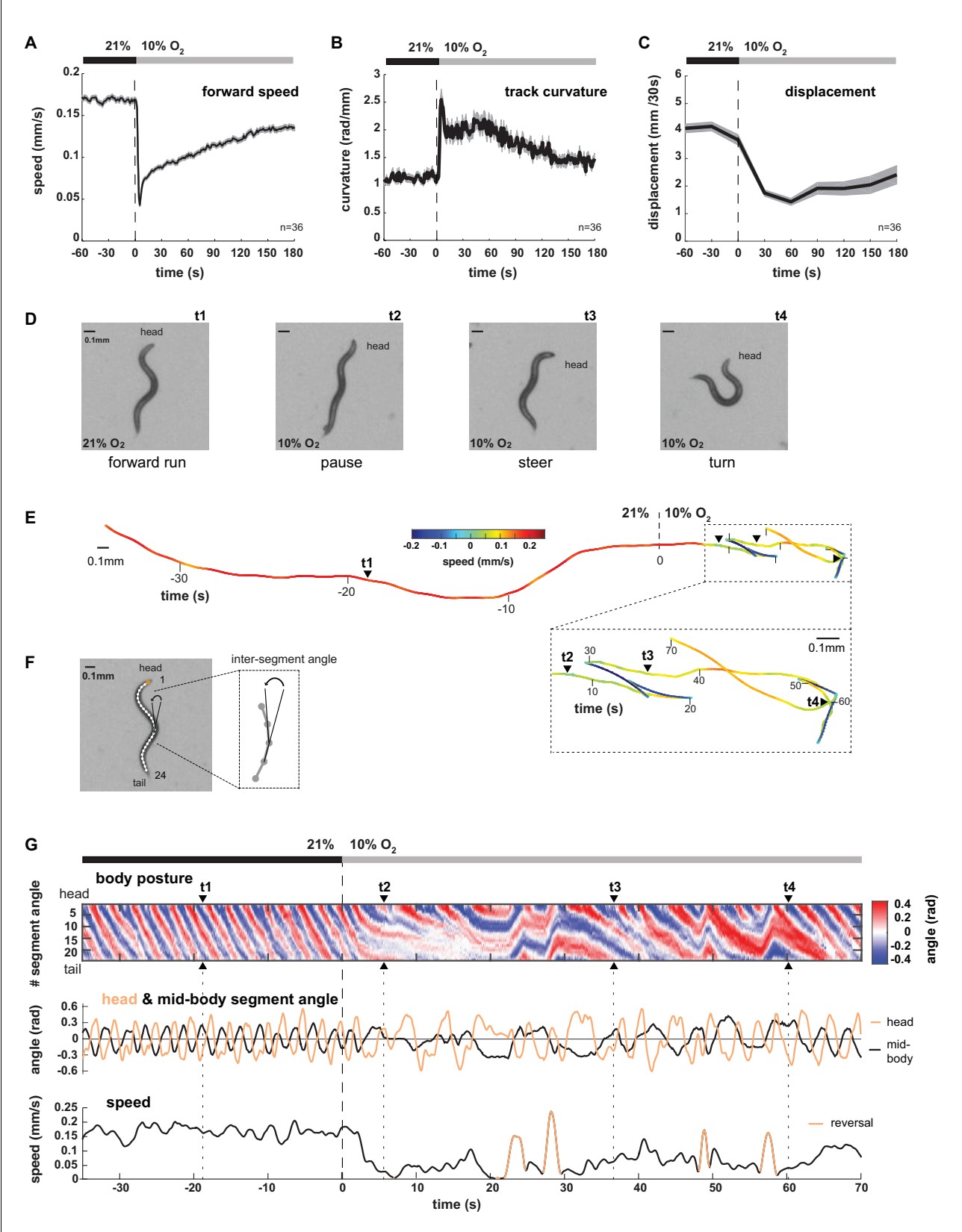

**Figure 1.** Worms transit from LDT to ARS after oxygen-sensory stimulation. (**A–C**) Locomotion parameters from population assays of wild type (N2) worms stimulated with $O_2$ downshifts from 21% to 10%. Traces show mean, shadings show SEM. n = 36 populations assays (~25 animals per assay). *Figure 1 continued on next page*

*Figure 1 continued*

Time is relative to $O_2$ downshift (A) Forward movement speed based on centroid coordinates (1 s binning). (B) Curvature of forward moving trajectories (1 s binning). (C) Centroid displacement between 30 s intervals. (D–G) Representative example of a wild type (N2) worm. t1-4 indicate corresponding time points. Time is relative to $O_2$ downshift. (D) Single video frames: (t1) forward run at 21% $O_2$, (t2) pause after $O_2$ downshift, (t3) shallow and (t4) deep turning maneuver. (E) Trajectory of centroid position for 35 s at 21% and 70 s at 10% $O_2$, respectively. Color indicates crawling speed; reverse movement is shown as negative speed. (F) Worm image overlaid with a skeleton. The inset shows how inter-segment angles were calculated. (G) Top: Body posture kymograph of 24 inter-segment angles from head (1) to tail (24). Middle: time course of representative head (#2) and mid-body angle (#11). Bottom: translational speed of the worm's centroid. Reverse movement is indicated by color.

The following figure supplement is available for figure 1:

**Figure supplement 1.** Behavioral transition from LDT to ARS after oxygen downshift depends on oxygen-sensory neurons BAG.

changes in ambient $O_2$ and that communicate with $O_2$ sensory neurons have been partially described (*Busch et al., 2012*; *Kato et al., 2015*; *Laurent et al., 2015*). However, the navigational strategies used during $O_2$ chemotaxis are uncharacterized.

In the present study we use $O_2$ chemotactic responses of food-deprived *C. elegans* as a paradigm to experimentally control a shift from LDT to ARS-like behavior. We show that the worm's undulatory motions can be decomposed into two modes corresponding to undulation and turning motions. By controlling the coordination of these modes animals can gradually adjust their locomotion strategy on a behavioral spectrum ranging from LDT to $O_2$-induced ARS. The interneurons AVK and DVA releasing the neuropeptides FLP-1 and NLP-12, respectively, are required for the control over this fine adjustment in response to acute sensory stimulation as well as during spatial navigation in $O_2$ gradients. In summary, this work shows how neuromodulatory circuit elements control elementary motor components to flexibly adjust the strategy of locomotion.

## Results

### A drop in ambient $O_2$ concentrations evokes a shift from LDT to $O_2$-induced ARS

We analyzed the locomotion behavior of 1 hr food-deprived worm populations filmed while freely crawling on agarose in a behavioral flow arena, which permitted tight control over ambient $O_2$ concentrations (*Zimmer et al., 2009*). An immediate drop from atmospheric 21% to 10% $O_2$ ($O_2$ downshift) evoked a variety of behavioral changes, including transient reduction of locomotion speed and concomitant up-regulation of reversals and omega turns as previously described (*Zimmer et al., 2009*) (*Figure 1A*, *Figure 1—figure supplement 1*). In addition, we observed up-regulation of shallow turning maneuvers during forward movement, evident as increased curvature of the animals' locomotion trajectories (track curvature) (*Figure 1B*). The compound effect of these behavioral changes was a drastic reduction in the spatial displacement of worms during the first three minutes upon $O_2$ downshift (*Figure 1C*). Thus, a sudden drop in environmental $O_2$ concentrations evoked a change in the locomotion strategy switching from predominantly forward crawling to a mixture of reduced crawling and increased reorientations, i.e. from LDT to $O_2$-induced ARS (see also *Video 1* and *Video 2* upper left panel). This interpretation is in accordance with previous literature classifying similar behavioral strategy changes in the context of removal from food (*Calhoun et al., 2015*; *Gray et al., 2005*; *Hills et al., 2004*; *Peliti et al., 2013*; *Tsalik and Hobert, 2003*; *Wakabayashi et al., 2004*). The changes in behavioral parameters all depended on an $O_2$ chemosensory neuron pair, termed BAG, which is activated upon $O_2$ downshift (*Zimmer et al., 2009*) (*Figure 1—figure supplement 1C*). We first qualitatively evaluated these behaviors by examining single worms (see *Video 1*). Upon $O_2$ downshift, animals tended to stop forward locomotion and adopted complex and variable postures, unlike the more regular sinusoid-like postures associated with LDT (*Figure 1D,E*). Some of these postures resembled omega turns, while others were associated with shallow turns and pauses (*Figure 1D*). In order to obtain quantifiable time-series of animal postures, we skeletonized the worm images and calculated 24 inter-segment angles for each worm and movie frame (*Figure 1F*); an example is shown by the kymograph in *Figure 1G*. LDT was associated with long stretches of regular and coherent body posture patterns, while $O_2$-induced ARS after $O_2$ downshift exhibited variability in the amplitude,

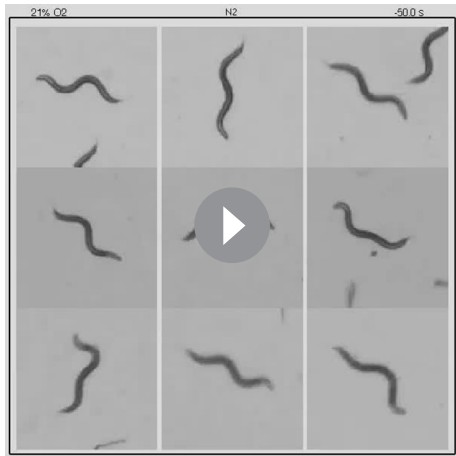

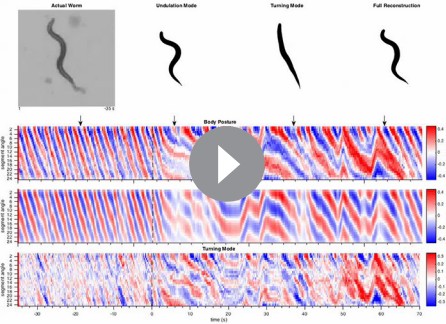

**Video 1.** Wild type N2 *C. elegans* behavioral responses to $O_2$ downshift. Exemplary wild type (N2) worms from population behavioral assays at 21% ambient $O_2$ and after a sudden shift at 10% $O_2$. Time is relative to the $O_2$ downshift. Note how the animals slow down their locomotory rate and up-regulate more complex and flexible postures. Time lapse 2x (20fps).

**Video 2.** Animated undulation and turning mode shapes of a wild type N2 worm. Reconstructed worm shapes of the decomposed undulation and turning modes and their superpositions from an exemplary wild type (N2) worm (same as in *Figures 1D–G*, *2C*, *D*). Time is relative to the $O_2$ downshift from 21% to 10%. Upper left panel shows original movie frame captures. The right panels show worm shapes reconstructed from undulation mode, turning mode and body posture time series shown in the kymographs below. The undulation mode corresponds to regular sinusoid-like worm shapes, whereas the turning mode corresponds to more strongly bent postures during shallow turns and omega turns. The diverse and complex postures observed are obtained by simple linear addition of undulation and turning shapes. Time lapse 1x (10fps).

direction and frequency of body postural changes (*Figure 1G*). Note that shallow turns, slowing bouts, reversals and omega turns also occurred sparsely during LDT; however, during $O_2$-induced ARS these behaviors were up-regulated 2-4-fold (compare pre- with post- stimulus period in all panels of *Figure 1*, *Figure 1—figure supplement 1*).

In summary, LDT is associated with extended periods of coherent movement across all body segments leading to efficient relocation, which are only sparsely interspersed by reorientation maneuvers, . Contrarily, $O_2$-induced ARS is associated with slow locomotion and frequent reorientation maneuvers mediated through more complex postures.

## Worm locomotion can be described as a linear composition of undulation and turning modes

Having observed different postural characteristics underlying LDT versus $O_2$-induced ARS, we aimed for a more detailed and precise description thereof. *C. elegans* uses 95 body wall muscles to move. However, the coordinated motions of the muscle-associated body segments can be quantified with far fewer than 95 parameters, leading to a reduced but comprehensive description of body posture (*Stephens et al., 2008*). Briefly, in this approach, principal components analysis (PCA) (*Jolliffe, 2002*) calculates the eigenvectors (termed eigenworms, EWs) of the covariance matrix of inter-segment angles observed over many posture time series (*Figure 2A*). EW's are a set of representative worm postures (as defined by a vector of inter-segment angles), in descending order of explanatory power; i.e. each EW contributes a certain amount of postural variance and linear combinations of the first four EWs account for the gross majority (>85%) of the observed postural variance (*Figure 2B*). For example, the first two eigenworms (EW1-2) capture the dominant correlations between segment body angles as reflected in *Figure 2A*, and subsequent EWs capture secondary correlational structure between segment body angles. We found that despite the strong effect of $O_2$ downshift on the animals' postures, the respective underlying EWs were nearly identical (*Figure 2A*). However, the relative contributions of each EW changed: the percentage of total variance explained by EW1 and EW2 decreased whereas the percentage of total variance explained by lower-order EWs increased (e.g. EW3-6, *Figure 2B*). Thus, the different postures observed during LDT versus $O_2$-induced ARS

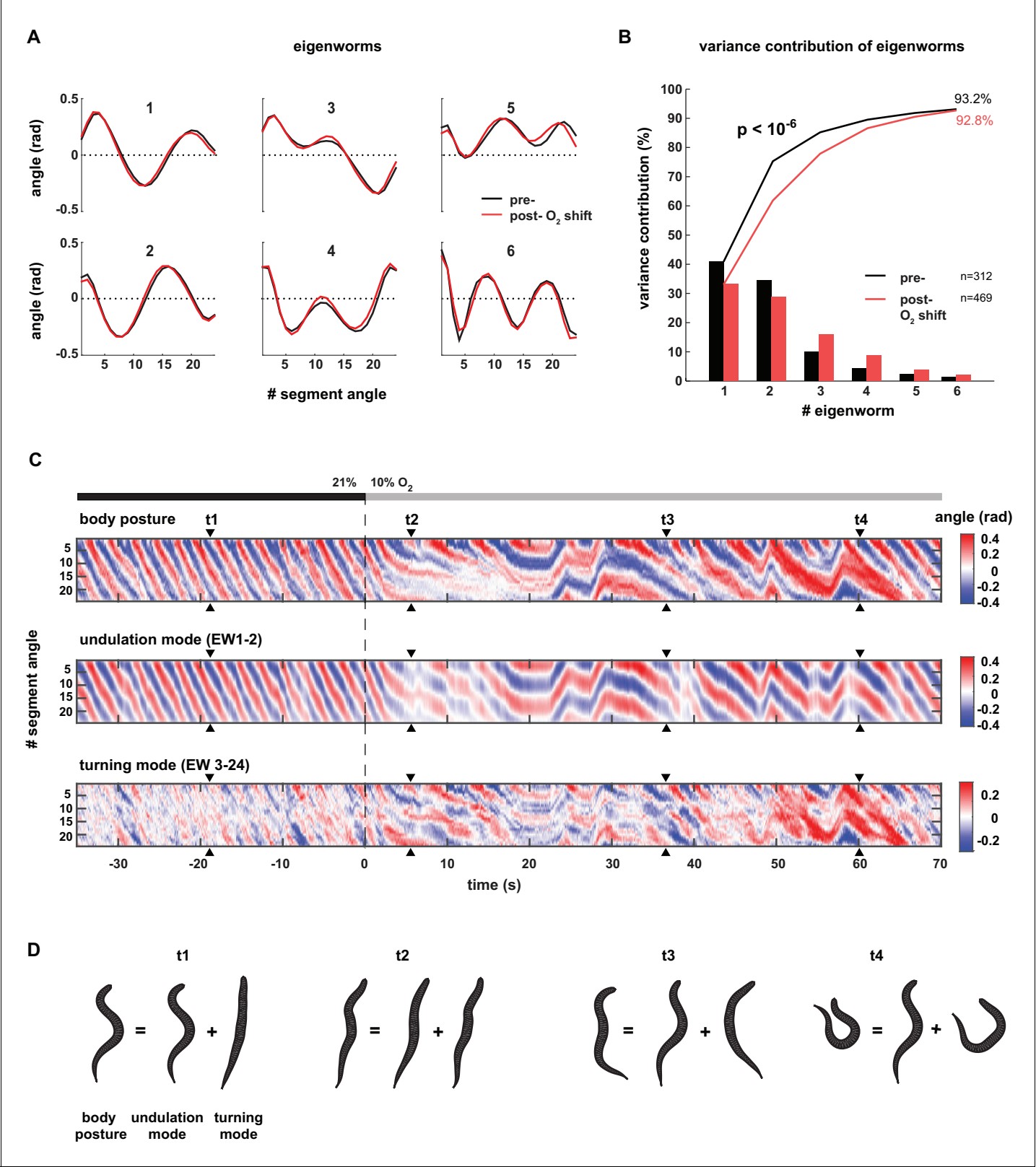

**Figure 2.** Decomposition of the body posture into undulation and turning motor patterns. (**A**) The panels show the top six of 24 eigenworms (EWs) calculated from pooled segment angle time series of wild type (N2) worms. Black and red EWs are calculated from 60s intervals pre (n = 312) or post- (n = 469) $O_2$ downshift respectively. (**B**) Bars show the variance explained by the EWs shown in A. Traces show cumulative values. The p-value was

*Figure 2 continued on next page*

Figure 2 continued

determined by a random resampling approach (see Materials and methods) and estimates an upper boundary of the probability that the difference between the cumulative variance distributions would occur by chance. (C) Example kymographs from the same worm shown in *Figure 1D–G*. Time is relative to $O_2$ downshift. Arrowheads indicate time points t1-4 as in *Figure 1*. Top: body posture as in *Figure 1G*. Middle: undulation mode reconstructed from EW 1–2. Bottom: turning mode reconstructed from EW 3–24. (D) Worm shapes graphically reconstructed from body posture, undulation mode and turning mode at time points t1-4. Note that the worm's posture is a linear sum of undulation and turning mode.

The following figure supplement is available for figure 2:

**Figure supplement 1.** Characterization of undulation and turning modes by signal cross-correlation analysis.

can be described by linear combinations of the same elementary shapes, the relative weights of which are modulated upon stimulation.

Next, we employed the eigenworm approach to obtain a more intuitive, yet quantitative, description of the worm postures characterizing LDT vs. $O_2$-induced ARS. Here and in all subsequent analyses throughout this manuscript we used canonical EWs calculated from the full wild type N2 time series (n = 36 population assays, $\sim$25 animals per assay). The combination of EW1 and 2 corresponds to the regular undulatory movements along the worm's body, which generates forward and backward crawling (*Stephens et al., 2008*). We thus reconstructed posture time series based only on EWs 1–2 (see Materials and methods for details). This led to a regular wave-like pattern corresponding to the undulatory worm movements; we termed this pattern the 'undulation mode' (see *Figure 2C* for an example). Subtracting the undulation mode from the full body posture time series, which is equivalent to reconstructing posture time series based on all remaining EWs 3–24, revealed a strikingly organized residual mode. Strong transients in the residual time series coincided with shallow turns as well as with highly bent omega turns (see *Figure 2C* for examples); we thus termed this time series the 'turning mode'. The turning mode was qualitatively different in comparison to the undulation mode: the head moved in anti-phase with the tail and in phase with the mid-body whereas the undulation mode was generated by anti-phase undulations of head and mid-body and in phase undulation of head and tail. We quantitatively assessed these differences by signal cross-correlation analysis (*Figure 2—figure supplement 1A–D*). The undulation mode and turning mode were not independent as they were often phase-locked: signal cross-correlation analysis showed that the onset of turning movements lagged on average $\sim$0.6 s behind the undulation mode (*Figure 2—figure supplement 1E,F*). We graphically synthesized worm shapes from the undulation mode and turning mode separately, which illustrated that the undulation mode corresponded to regular sinusoid-like worm shapes, whereas the turning mode corresponded to more strongly bent postures during shallow turns and omega turns. The diverse and complex postures observed in worms were obtained by simple linear addition of undulation and turning shapes (*Figure 2D*; *Video 2*).

In conclusion, apparently complex postures of *C. elegans* can be linearly composed out of simpler elementary shapes, which constitute behaviorally relevant motor patterns of the worm: the undulation mode describes regular forward and backward crawling, onto which the turning mode imposes turning movements.

## Turning mode and undulation mode are reciprocally regulated

We used the eigenworm decomposition method to calculate three instantaneous (i.e. for each movie frame) parameters of worm locomotion: (1) undulation amplitude, a measure of the curvature across the body contributing to undulation movements, calculated as the sum of the absolute values of all inter-segment angles of the undulation mode, (2) undulation frequency of the undulation movements, determined from the projection amplitude time series of EW1 and EW2 (*Stephens et al., 2008*) (see Materials and methods for details), and (3) turning amplitude, a measure of how strong the animals bend their bodies in order to turn, calculated as the sum of the absolute values of all inter-segment angles of the turning mode. Biomechanical calculations indicate that both body bending amplitude and frequency synergistically generate thrust against the substrate thereby contributing to the animals' locomotion speed (*Gray, 1953*). The sum of the absolute values of all inter-segment angles of the body posture is a measure of the total curvature (or tortuousness) of the

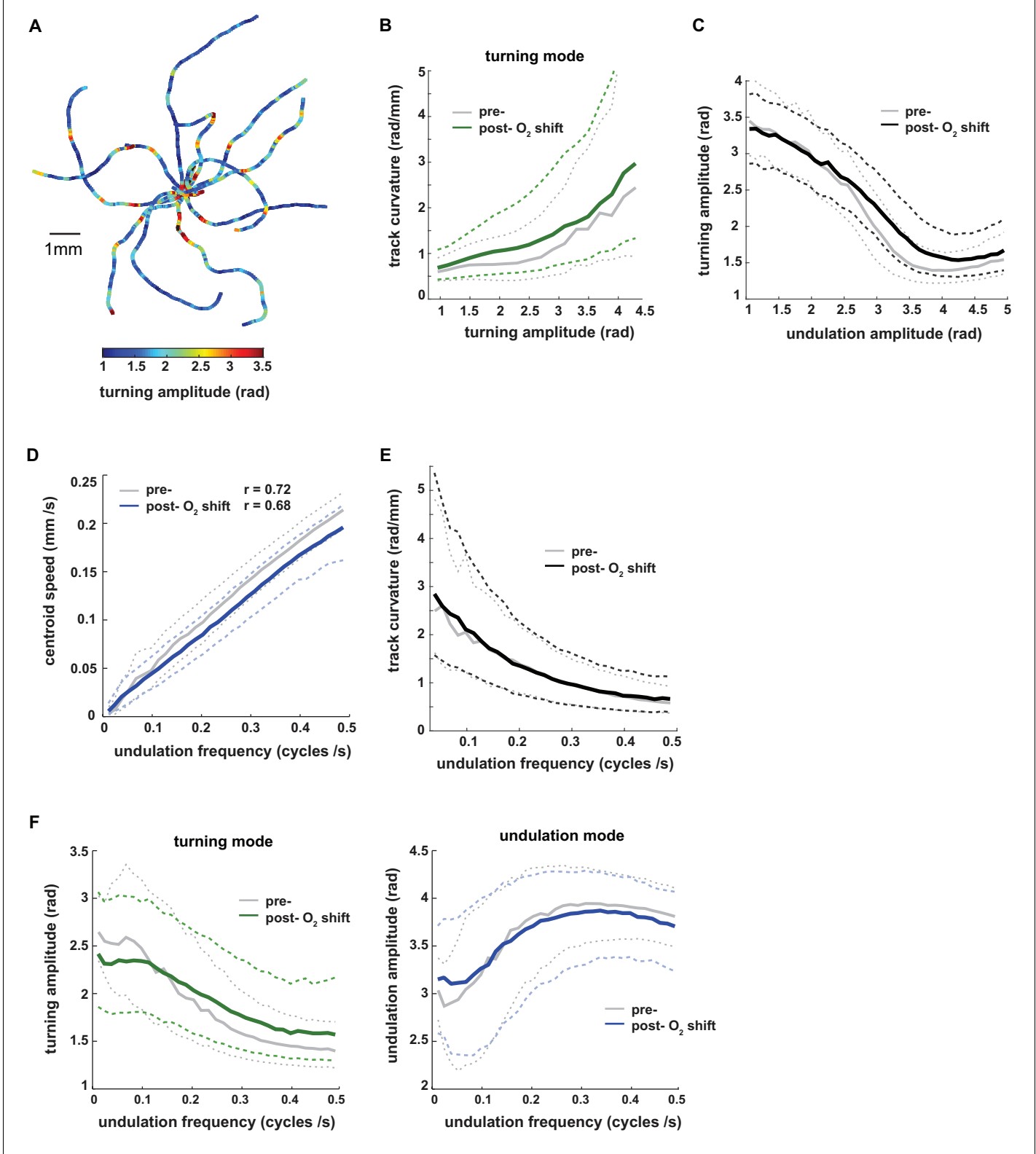

**Figure 3.** Turning amplitude is reciprocally regulated with undulation amplitude and undulation frequency. (A) Exemplary worm trajectories during the 10% $O_2$ interval aligned at their starting points with color code indicating turning amplitude. (B–F) The graphs show interdependencies of the indicated locomotion parameters during forward movement. Solid lines show the medians and dashed lines show interquartile ranges on the y–axis. Data are binned by the corresponding values on the x-axis. Tick marks indicate bin medians of the x-axis. Data from wild type N2 worms (n = 36 assays, ~25

*Figure 3 continued on next page*

*Figure 3 continued*

animals per assay) during 3 min intervals pre- (gray) and post- (black or color) $O_2$ downshift were analyzed. Very high values on the x-axis are omitted due to data sparseness. (B) Track curvature versus turning amplitude. Bin width = 0.2 rad. (C) Turning amplitude versus undulation amplitude. Bin width = 0.1 rad. (D) Centroid speed versus undulation frequency. Bin width = 0.015 cycles/s. Pearson correlation coefficients are indicated. (E) Track curvature versus undulation frequency. Bin width = 0.015 cycles/s. (F) Turning amplitude (left) or undulation amplitude (right) versus undulation frequency. Bin width = 0.015 cycles/s.

animals' posture, which we term body amplitude. To obtain separate measures of worm locomotion during LDT and $O_2$-induced ARS, we performed analyses on the pre- and post-stimulus periods. High signals in turning amplitude consistently coincided with regions of high track curvature, i.e. events when animals adjusted their heading direction (see *Figure 3A* for examples); track curvature was a graded function of turning amplitude (*Figure 3B*). Turning amplitude and undulation amplitude exhibited a graded inverse relationship to each other (*Figure 3C*). As expected, undulation frequency exhibited a tight linear relationship with translational locomotion speed (*Figure 3D*) (See also reference [*Stephens et al., 2008*]). We found that directional changes during forward movement were associated with slow locomotion: there was an inverse relationship between undulation frequency and centroid track curvature (*Figure 3E*). High undulation amplitude was associated with fast locomotion (high undulation frequency) whereas high turning amplitude was associated with slow, low-frequency locomotion (*Figures 3F*). All relationships described were preserved during both pre- and post- stimulus periods (*Figure 3B–F*).

In conclusion, turning amplitude, undulation amplitude and undulation frequency are coupled: during turning maneuvers, animals reduce the amplitude and speed of undulation motions, leading to slow locomotion. Contrarily during periods of fast locomotion turning maneuvers are suppressed. Under our experimental conditions, the coupling of these motor parameters is a robust property of worm locomotion during both spontaneous and stimulus-evoked behaviors.

## Undulation and turning modes are gradually regulated by sensory input

We next analyzed the averaged time courses of the turning and undulation amplitudes in response to the stimulus. Upon stimulus onset, the mean amplitudes were abruptly up- or down-regulated, respectively, and gradually returned to baseline within three minutes (*Figure 4A*). For comparison, we calculated the time course of total body amplitude, which changed upon stimulation with a biphasic response profile, i.e. initial suppression followed by transient up-regulation (*Figure 4A*).

Based on the graded relationships exhibited by turning and undulation parameters (*Figure 3B–F*) we hypothesized that animals were able to gradually control the contributions of undulation and turning motions. To test this hypothesis, we stimulated the animals with a temporal oxygen ramp ranging from atmospheric 21% to 4% $O_2$ (-0.01%/s). This protocol caused a subtle change in mean total body amplitude but stronger and gradual reciprocal changes in mean undulation and turning amplitudes (*Figure 4B*). We then plotted the fraction of postures with a given turning amplitude as distribution functions during subsequent time segments of the ramp. We did not observe bimodality but instead that the skew of these distributions gradually increased towards higher levels (*Figure 4C*). Thus, animals were able to gradually adjust turning and undulation motions within a wide continuous range, as further revealed in a two-dimensional density plot of turning amplitude against undulation frequency (*Figure 4D*).

In summary, oxygen stimulation results in a shift in the contributions of undulation and turning motor patterns. Animals display control of the strength of undulation and turning motions in response to both abruptly and gradually changing environments. LDT and $O_2$-induced ARS behaviors appear to lie on a continuum of behavioral mode compositions.

## Neural control over undulation and turning modes through antagonistic peptidergic interneurons

We hypothesized that undulation and turning modes are under control of some of the animals' premotor interneuronal circuits. To identify such circuits we analyzed candidate neuropeptide modulators that were reported to affect body posture and found roles for FLP-1 and NLP-12 neuropeptides: *flp-1* mutants have been reported to exhibit elevated body curvature and locomotion speed

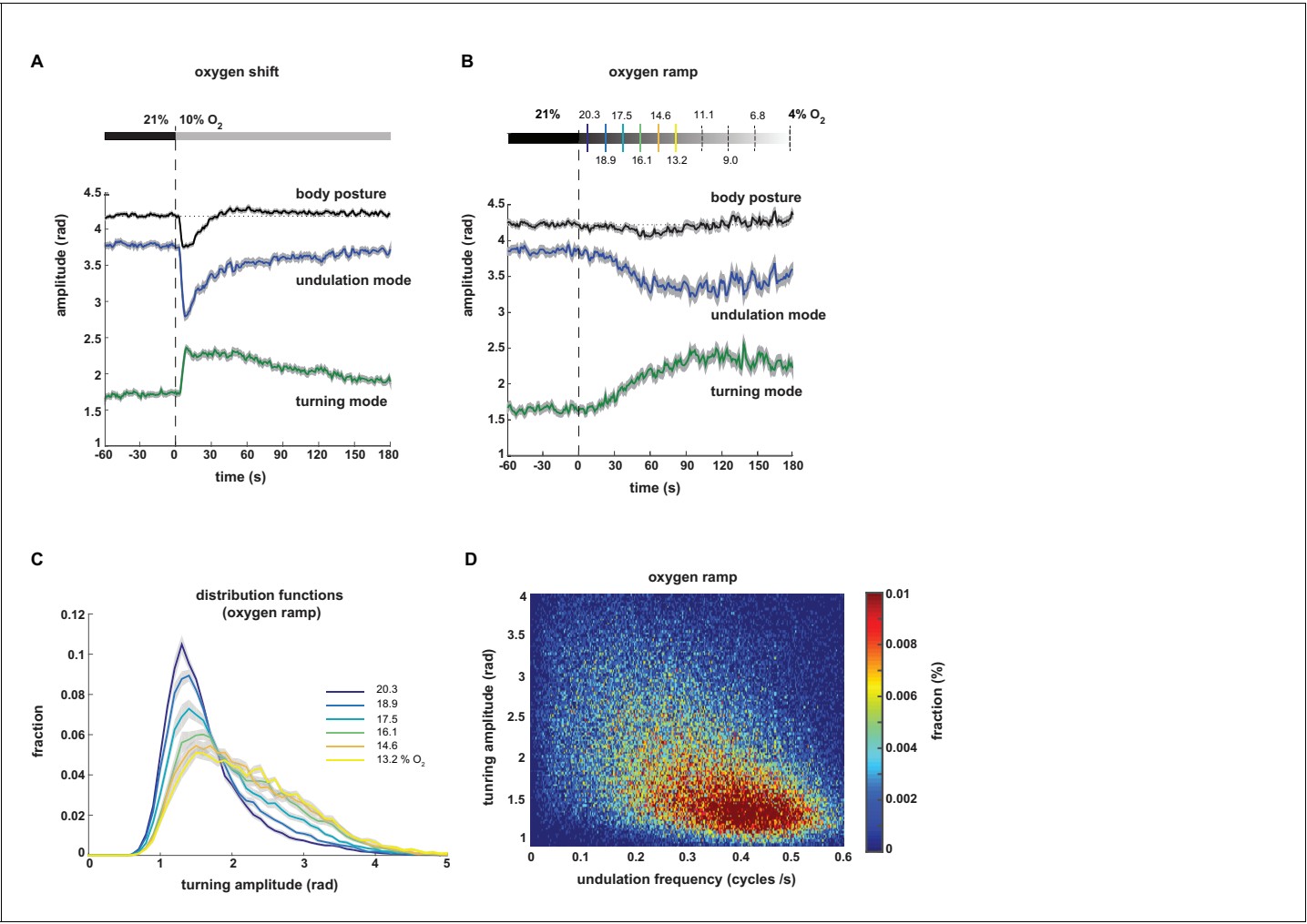

**Figure 4.** Undulation amplitude and turning amplitude are inversely and gradually regulated in response to $O_2$ stimulation. (A,B) Behavioral responses of wild type N2 worms during forward movement evoked by decreasing $O_2$ concentrations. Traces show mean amplitude of body posture (black), undulation mode (blue) and turning mode (green); shadings indicate SEM. Time is relative to onset of $O_2$ concentration change. (A) $O_2$ downshift from 21% to 10% $O_2$ (n = 36 assays, ~25 animals each) and (B) gradual $O_2$ ramp from 21 to 4% $O_2$ (n=15 assays, ~25 animals each). Some $O_2$ concentrations during the ramp are indicated on top. Colors correspond to (C). (C) Trial-mean (± SEM) fractional distribution functions of turning amplitude from the same data shown in (B). All distributions are calculated from consecutive 15 s intervals during the ramp with amplitude bin width of 0.1 rad. The legend indicates the median $O_2$ concentration during each interval. Color code corresponds to (B). (D) 2D density map showing the fractional distribution of forward undulation frequency versus turning amplitude from all worms (n ≈ 375) during the ramp assays shown in (B, C). Data from -60 s to 180 s were included. Bin width are 0.002 cycles/s and 0.025 rad. The y- and x-axes omit very high values due to data sparseness.

(*Chang et al., 2015*; *Nelson et al., 1998*). NLP-12 neuropeptides have been reported to be exclusively released from the proprioceptive interneuron DVA; interfering with the DVA/NLP-12 system leads to decreased body curvature and locomotion speed (*Bhattacharya et al., 2014*; *Garrison et al., 2012*; *Hu et al., 2011*; *Janssen et al., 2008*; *Li et al., 2006*). We first confirmed these previously reported results in our 1 hr food deprivation paradigm. We assayed *flp-1* and *nlp-12* mutants, DVA ablated (DVA-) animals, as well as transgenic *nlp-12* rescue strains in DVA, and found that all phenotypes could be recapitulated in 1 hr fasted worms (*Figure 5A,B*; *Figure 6A*). Next, we aimed to identify the relevant site of FLP-1 release. *flp-1* is expressed in various interneurons of the worm's head and ventral ganglia (*Nelson et al., 1998*). We found that *flp-1* expression in the single ventral ganglion interneuron class AVK, which we observed to be the neuron class with strongest expression, is sufficient to restore a wild type phenotype (*Figure 6A*). We generated AVK-animals, which caused enhanced body posture amplitude similar to *flp-1* mutants (*Figure 5A,B*). *flp-1;nlp-12* double mutants as well as AVK-;DVA- animals exhibited intermediate total body amplitude

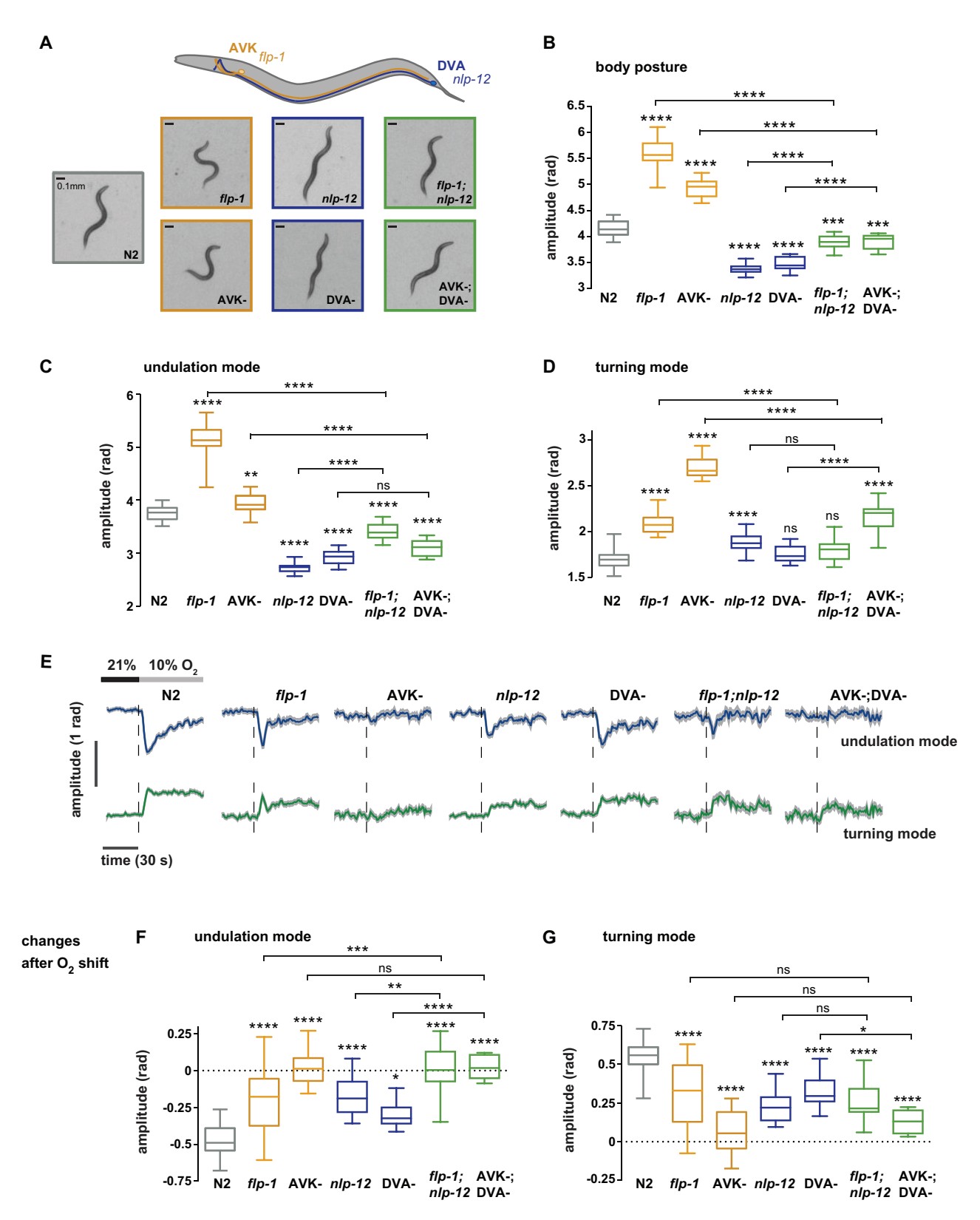

**Figure 5.** Regulation of undulation mode and turning mode through antagonistic peptidergic interneurons. (**A**) Top: Anatomical sketch of interneurons AVK and DVA. All other panels show representative video frames during forward movement at 21% $O_2$ of worms with indicated genotypes. *flp-1* and *Figure 5 continued on next page*

*Figure 5 continued*

*nlp-12* are loss of function mutations. AVK- and DVA- denote transgenes driving caspase expression leading to cell death in the respective neuron. Note the effects of genotypes on body posture. Heads are pointing upwards. (**B–D**) Boxplots of amplitudes during forward movement measured during a 4 min interval at 21% $O_2$ of the (**B**) body posture, (**C**) undulation mode and (**D**) turning mode. (**E**) Time profiles of mean undulation amplitude (blue) and mean turning amplitude (green). Shadings show SEM. Dashed lines indicate onset of $O_2$ downshift. The traces are vertically aligned to emphasize differences in behavioral responses. The vertical bar indicates amplitude; the horizontal bar indicates time axis. (**F, G**) Boxplots showing changes of (**F**) undulation amplitude and (**G**) turning amplitude in response to the $O_2$ downshift. Changes were calculated by subtracting the means of 60s intervals post-downshift from the means of 60s pre-downshift intervals. Boxplots in all panels display trial-median and interquartile range with 5–95 percentile whiskers. Asterisks on top of box plots indicate significance levels compared to wild type N2 and asterisks on top of brackets indicate significance levels for selected comparisons, using one-way-ANOVA with Sidak's correction for multiple comparisons (****$p \leq 0.0001$, ***$0.0001 < p \leq 0.001$, **$0.001 < p \leq 0.01$, *$0.01 < p \leq 0.05$, ns $p > 0.05$). Number of experiments (~25 animals per assay): N2 n = 36, *flp-1* n = 29, AVK- n = 21, *nlp-12* n = 21, DVA- n = 13, *flp-1;nlp-12* n = 12 and AVK-;DVA- n = 10.

The following figure supplements are available for figure 5:

**Figure supplement 1.** Postures of manipulated animals with mis-regulated motor patterns largely lie within the wild type N2 posture space.

**Figure supplement 2.** Regulation of locomotion speed through antagonistic peptidergic interneurons.

**Figure supplement 3.** Gradual behavioral transitions depend on *flp-1* and *nlp-12* neuropeptide genes.

similar to wild type levels, indicating an antagonism between both systems (***Figure 5A,B***). For comparison across different genetic backgrounds, we calculated undulation and turning modes for all genetic backgrounds using the wild type N2 eigenworms as a common basis. The decomposition revealed that a large proportion of altered body posture across all phenotypes could be explained by a corresponding effect on undulation amplitude. Interfering with AVK/FLP-1 function increased undulation amplitude, while interfering with DVA/NLP-12 decreased undulation amplitude . *flp-1;nlp-12* double mutants exhibited intermediate phenotypes (***Figure 5C***). In addition, all genotypes except *flp-1;nlp-12* double mutants and DVA- animals showed an increase in turning amplitude, which was especially strong for *flp-1* mutants and animals lacking AVK (***Figure 5D***). A comparison of 2D probability density distributions of undulation vs. turning mode revealed that to a large extent the postures of mutants and manipulated strains reflected postures that were exhibited, however less frequently, by wild type animals (***Figure 5—figure supplement 1***). Besides their effects on worm postures at constant atmospheric $O_2$ levels, all analyzed mutant and cell ablation strains were compromised in regulating undulation and turning mode in response to $O_2$ downshift. The strongest defects in stimulus-evoked postural changes were seen in strains lacking AVK neurons, as well as in *flp-1;nlp-12* double mutants (***Figure 5E–G***). The DVA/NLP-12 and AVK/FLP-1 systems were both implicated in controlling locomotion speed (***Figure 5—figure supplement 2***). Cell-specific expression of *nlp-12* in DVA and *flp-1* in AVK partially restored the respective phenotypes, confirming these cells as important release sites for the respective neuropeptides (***Figure 6A–F***; ***Figure 5—figure supplement 2C,E***).

Next we tested whether the DVA/NLP-12 and AVK/FLP-1 systems were required for fine control of behavior in the $O_2$ ramp stimulus paradigm. We focused on *flp-1;nlp-12* double mutants, since their baseline behavioral parameters during LDT were similar to wild type N2 animals (***Figure 5B–D***). However, we hypothesized that if the neuropeptides were part of a control system of worm postures, the double mutants should lack control over the composition of behavioral modes. Indeed, in the ramp paradigm the mutant animals' modulation of undulation and turning amplitudes was largely compromised (***Figure 5—figure supplement 3***). This phenotype could be rescued by cell-specific expression of *nlp-12* in DVA and *flp-1* in AVK (***Figure 6G–H***).

In conclusion, control of the composition of undulation and turning modes can be partially dissected at the genetic and single neuron level. The DVA/NLP-12 and AVK/FLP-1 systems are required for normal body posture during LDT in constant environmental conditions as well as for the differential fine control of undulation and turning motions in response to both abruptly and smoothly changing environments. Our data suggest that decomposition of posture into undulation and turning modes is not only a useful quantitative description of behavior, but that these motor patterns are under differential neural control.

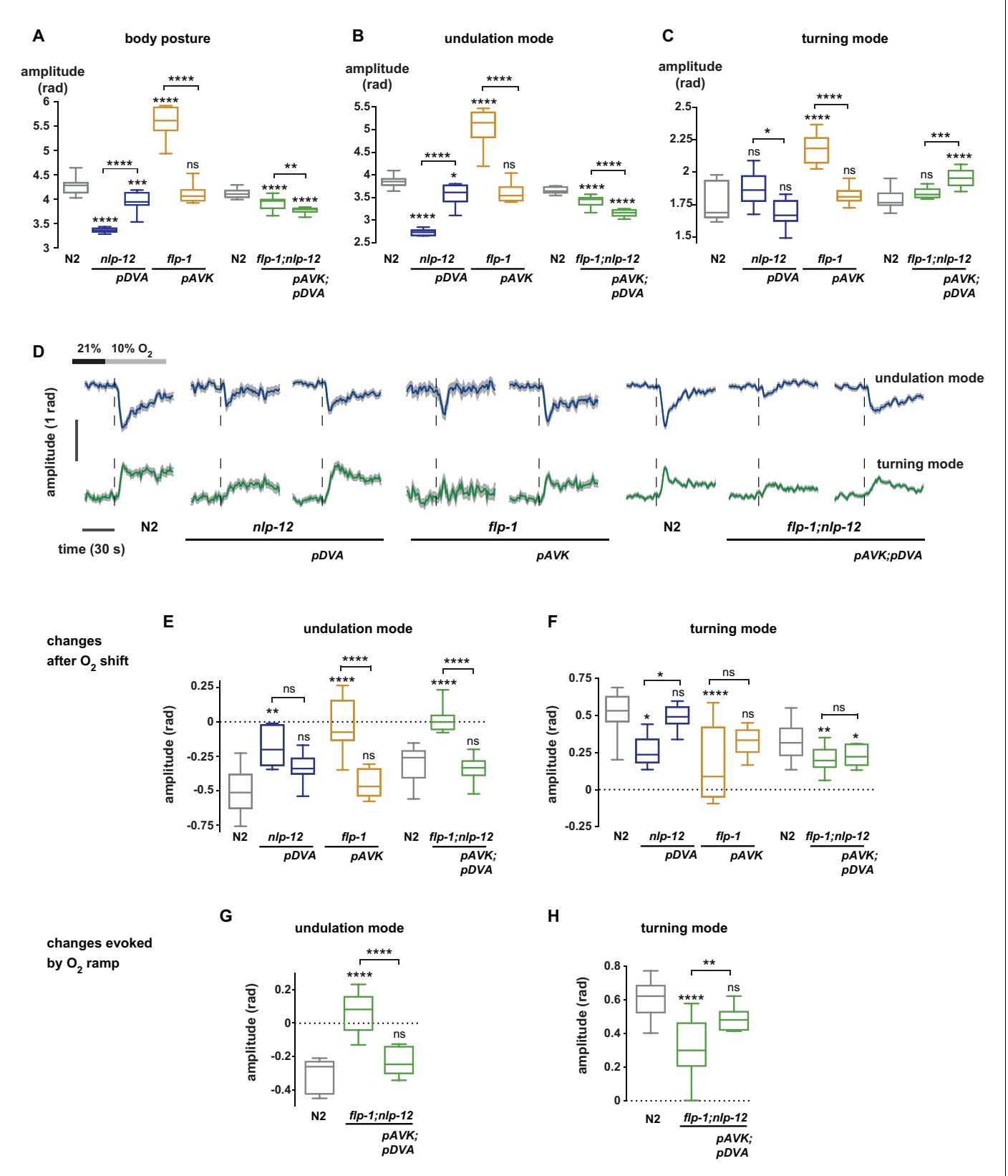

**Figure 6.** Peptidergic regulation of undulation and turning mode by *flp-1* and *nlp-12* through interneurons AVK and DVA. (A–C) Boxplots of amplitude during forward movement measured during a 4 min interval at 21% $O_2$ of the (A) body posture (B) undulation mode and (C) turning mode. (D) Time

*Figure 6 continued on next page*

Figure 6 continued

profiles of mean undulation amplitude (blue) and mean turning amplitude (green). Shadings show SEM. Dashed lines indicate onset of $O_2$ downshift. The traces are vertically aligned to emphasize differences in behavioral responses. The vertical bar indicates amplitude; the horizontal bar indicates time axis. (E–H) Boxplots showing changes of undulation amplitude (E, G) and turning amplitude (F, H) in response to the $O_2$ downshift (E, F) or to an $O_2$ ramp (G, H). Changes were calculated by subtracting the means of 60 s intervals post-downshift from the means of 60 s pre-downshift intervals (E, F) or by subtracting the means of 60 s intervals at the end of the ramp from the means of 60 s intervals before ramp onset (G, H). Genotypes indicate transgenic rescue using cell specific promoters (pDVA, pAVK). Boxplots in all panels display trial-median, interquartile range and 5–95 percentile whiskers. Asterisks on top of box plots indicate significance levels compared to wild type N2 and asterisks on top of brackets indicate significance levels for comparisons with the respective mutant, using one-way-ANOVA with Sidak's correction for multiple comparisons (****$p \leq 0.0001$, ***$0.0001 < p \leq 0.001$, **$0.001 < p \leq 0.01$, *$0.01 < p \leq 0.05$, ns $p > 0.05$). Number of experiments ($\sim$25 animals per assay): N2 n = 10, nlp-12 n = 6, pDVA: nlp-12 rescue under Pnlp-12 n = 11, flp-1 n = 11, pAVK: flp-1 rescue under Pflp-1 fragment n = 9, N2 n = 11, flp-1;nlp-12 n = 11, pAVK;pDVA: flp-1 rescue under Pflp-1 fragment & nlp-12 rescue under Pnlp-12 n = 11.

## AVK neuronal activity correlates with locomotion speed and repression of turning

In order to gain more mechanistic insight into the action of DVA and AVK neurons, we investigated the relationship between their activity and behavior. We performed high-magnification (40x) fluorescence $Ca^{2+}$ imaging, using the indicator GCaMP5K, simultaneously with lower magnification infrared behavioral recording of worms freely moving on agarose. A closed-loop tracking system kept neurons of interest in the field of view of the imaging objective (*Faumont et al., 2011*; *Kato et al., 2015*) and oxygen levels were controlled via a custom-fabricated oxygen flow arena mounted on the microscope stage (*Kato et al., 2015*). An example recording of AVK activity is shown in *Figure 7A*. We observed frequent $Ca^{2+}$ transients of varying magnitude, which were associated with changes in the animal's locomotion speed. Consistent with this observation, AVK $Ca^{2+}$ signals in average decreased upon $O_2$ downshift (*Figure 7B–C*). When measuring AVK activity in immobilized animals, we did not detect $O_2$ downshift-evoked activity changes (*Figure 7C*). In freely moving animals, we found that AVK activity positively correlated with locomotion speed (*Figure 7D*), but unlike previously reported speed-encoding interneurons, which exclusively encode either forward or reverse locomotion speed (*Kato et al., 2015*; *Li et al., 2014*), AVK correlated with speed during both forward and reverse movement (*Figure 7E*). Consistent with our findings that speed and turning motions were reciprocally regulated, AVK activity negatively correlated with turning amplitude during forward movement (*Figure 7F*). The relationship of AVK activity with behavioral output was unchanged when comparing pre- and post-stimulus periods (*Figure 7G–H*). Our activity recordings were corrected for motion artifacts by co-expressing mCherry and calculating GCaMP/mCherry fluorescence ratios; additional control experiments using $Ca^{2+}$-insensitive GFP confirmed that our measurements were GCaMP-specific and thus not derived from possible motion artifacts (*Figure 7—figure supplement 1*).

In summary, elevated AVK neuronal activity reflects high locomotion speed and low turning amplitude, thus corresponding to aspects of motion rather than to sensory inputs; similar observations have been made for many interneurons in *C. elegans* (*Kato et al., 2015*; *Larsch et al., 2013*; *Laurent et al., 2015*; *Li et al., 2014*; *Luo et al., 2014b*). The neural activity is consistent with the behavioral phenotypes of AVK- animals, which exhibit reduced locomotion speed (*Figure 5—figure supplement 2A,B*) and strong up-regulation of turning maneuvers (*Figure 5D*). Taken together, these results suggest that AVK activity promotes locomotion speed and suppresses turning. However, since AVK activity changes are low in immobilized worms we propose that AVK can influence these functions only in the context of feedback from the motor system.

## DVA neuronal activity correlates with multiple behavioral features including reversing, pausing and undulation amplitude

Next we tested the neural response of the DVA neuron in respect to behavioral output. *Figure 8A* shows an example trace of DVA neuronal activity in a freely moving worm. DVA $Ca^{2+}$ signals increased on average upon $O_2$ downshift (*Figure 8B,C* upper panel, see *Figure 8—figure supplement 1A,B* for motion artifact controls). We found that DVA activity correlated with multiple aspects of locomotory behavior: DVA activity rose on average at the transition from forward to backward

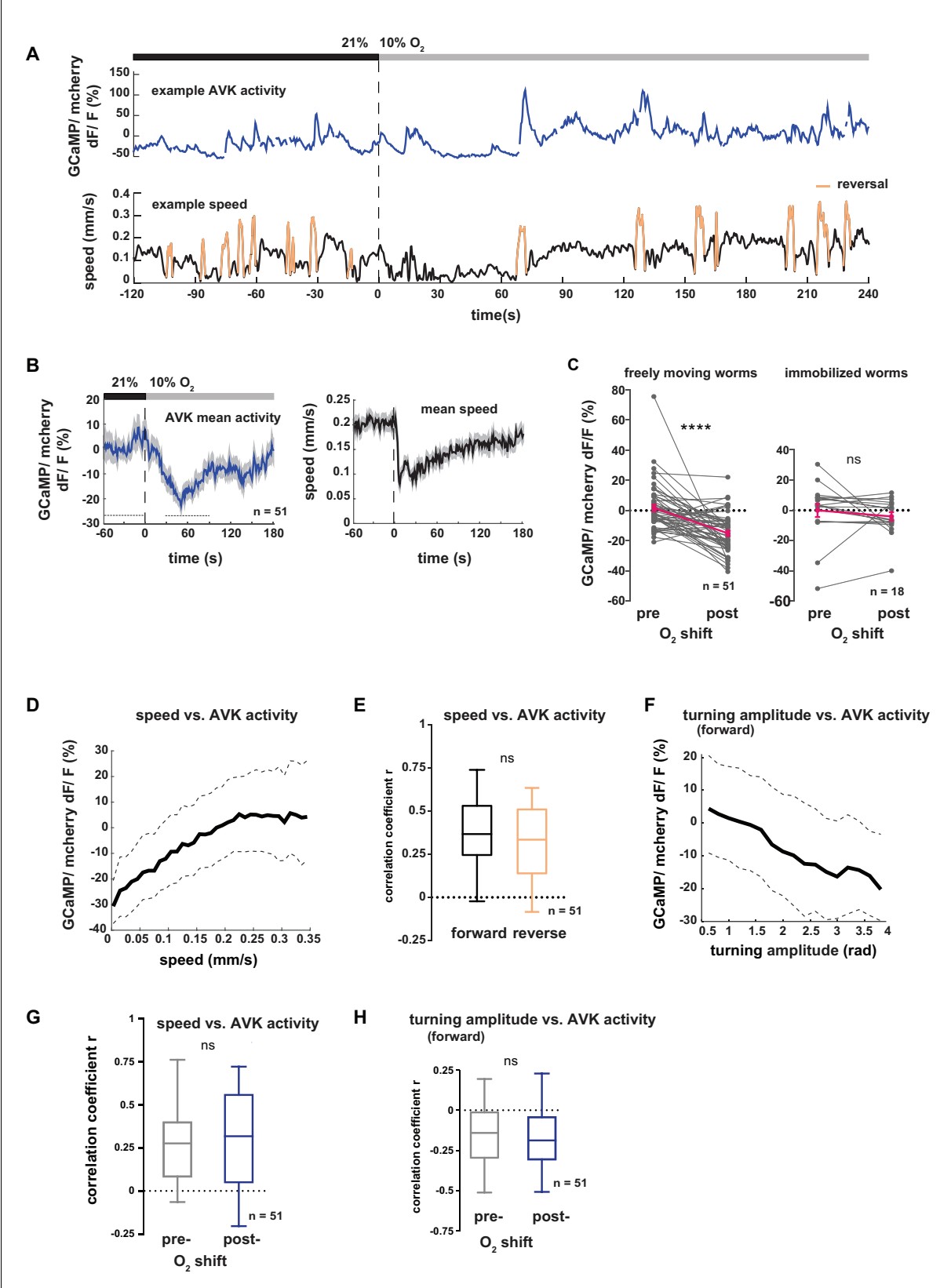

**Figure 7.** Neural activity of interneuron AVK reflects locomotion speed and suppression of turning motions. (A–H) Neural activity of AVK neurons was measured in freely moving worms by calcium imaging and is displayed as normalized ratio of GCaMP5K/ mCherry fluorescence. (A) Exemplary

*Figure 7 continued on next page*

*Figure 7 continued*

recording of neural activity (top) and locomotion speed (bottom). Time is relative to $O_2$ downshift. Periods of reverse movement are labeled in orange. (B) Mean AVK activity (left) and mean locomotion speed (right). Shadings indicate SEM. Time is relative to $O_2$ downshift. n = 51 worms. (C) AVK neural activity changes in response to $O_2$ downshift compared between freely moving worms and worms immobilized in a microfluidic device. Quantifications of mean activity during 60 s intervals pre- and post- $O_2$ downshift indicated by dashed lines in (B). Values from single recordings are shown in gray and population means ± SEM are shown in magenta. AVK activity significantly decreased in freely moving animals, but did not change in immobilized worms (****p≤0.0001; ns, p=0.34, paired t-test). (D) Median (solid line) and interquartile range (dashed lines) of AVK activity versus binned locomotion speed (bin size = 0.01 mm/s). (E) Box plots of linear correlation coefficients between forward or reverse movement speed and AVK activity calculated for each recording (n = 51). Unpaired t-test shows no significant difference (ns, p=0.13). (F) Median (solid line) and interquartile range (dashed lines) of AVK activity during forward movement versus binned turning amplitude (bin size = 0.2 rad). (G,H) Box plots of linear correlation coefficients calculated from time intervals pre- and post- $O_2$ downshift (n = 51 each). Paired t-tests show no significant differences. (G) Locomotion speed versus AVK activity; 2 min intervals (ns, p = 0.89). (H) Turning amplitude during forward movement versus AVK activity, 4 min intervals (ns, p=0.56). All boxplots display median, interquartile range and 5–95 percentile whiskers.

The following figure supplement is available for figure 7:

**Figure supplement 1.** Neural activity changes of interneuron AVK are not derived from motion artifacts.

directed movement (*Figure 8—figure supplement 1C–E*) and DVA was strongly activated during periods of paused locomotion (*Figure 8—figure supplement 1F, G*), a behavior that was observed more frequently upon $O_2$ downshift (*Figure 8—figure supplement 1H*). After restricting the analysis exclusively to periods of forward locomotion, i.e. removing reversal and pausing periods from the data, we did not detect any $O_2$ downshift-evoked change in the residual DVA activity (*Figure 8C*, lower panel). In contrast to AVK, a global correlation analysis did not detect a relationship between DVA activity and forward locomotion speed, turning amplitude or undulation amplitude ($R$ = -0.04, $R$ = 0.09, $R$ = 0.06, respectively). However, discernable DVA $Ca^{2+}$ transients were evident during periods of forward locomotion (*Figure 8A*). To test whether these signals had a behavioral correlate, we calculated $Ca^{2+}$ peak-triggered averages of undulation amplitude, turning amplitude and body posture amplitude. This approach showed that DVA $Ca^{2+}$ signals were on average associated with transient increases in undulation amplitude and transient decreases in turning amplitude (*Figure 8D*). Triggering body posture amplitude to DVA activity peaks did not reveal any discernible signal (*Figure 8—figure supplement 1I*); these findings highlight the strength of the eigenworm decomposition approach for decoding behavior from neural activity.

In summary DVA activity correlates to multiple aspects of behavior including the transition from forward to reverse movement and prolonged pausing. Consistent with the role of DVA/NLP-12 in promoting undulation amplitude (*Figure 5C*) DVA neural activity peaks co-occur with undulation amplitude increases and reciprocal turning amplitude decreases.

## Multiple behavioral strategies contribute to $O_2$ chemotaxis

In the previous sections we have described that worms coordinate a variety of locomotion parameters when responding to rapidly and gradually changing ambient $O_2$ concentrations. In the following sections we address whether animals make use of this behavioral repertoire in a navigational chemotaxis paradigm. Previous work has shown that *C. elegans* navigates $O_2$ gradients towards preferred intermediate $O_2$ concentrations avoiding both hypoxia and atmospheric $O_2$ levels (*Cheung et al., 2005*; *Gray et al., 2004*). $O_2$ sensing by BAG sensory neurons is required for these behaviors under fasted conditions (*Zimmer et al., 2009*). We thus analyzed the distribution of animal populations exposed to linear gas phase $O_2$ gradients ranging from 4% to 21% (0.6% $O_2$ /mm) generated in a previously reported microdevice (*Gray et al., 2004*) (*Figure 9A*). The conditions used in the present study excluded the hypoxia regime (<4% $O_2$) since the neuronal mechanisms of acute hypoxia avoidance in *C. elegans* have not been identified yet (*Gray et al., 2004*; *Zimmer et al., 2009*). The average distribution profile suggested that 1–1.5 hr food-deprived animals strongly avoided low $O_2$ concentrations, accumulated at intermediate $O_2$ concentrations around a bin center of 16.4% and slightly avoided atmospheric levels of 21% $O_2$ (*Figure 9B*). For each experiment a paired control experiment was performed (21% $O_2$ flow from both microdevice inlets), during which animals tended to slightly accumulate in the middle part of the arena (*Figure 9B*).

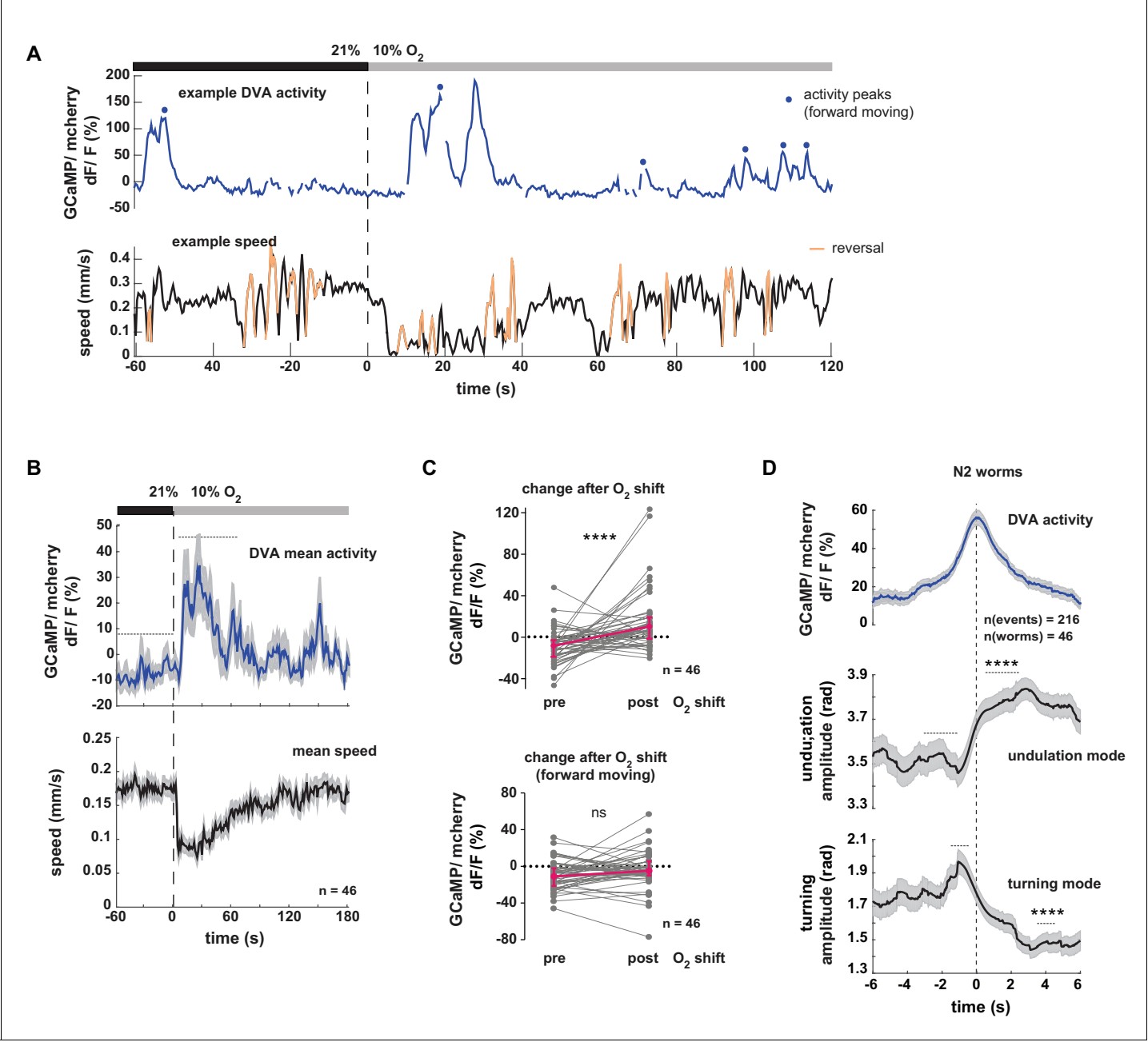

**Figure 8.** Neural activity of interneuron DVA is associated with the amplitude of undulation motions. (**A–D**) Neural activity of DVA neurons was measured by calcium imaging in freely moving animals and is displayed as normalized ratio of GCaMP5K/mCherry fluorescence. (**A**) Exemplary recording of neural activity (top) and locomotion speed (bottom). Time is relative to $O_2$ downshift. Periods of reverse movement are labeled in orange. Peaks in DVA calcium transient during forward displacement are marked by blue dots. (**B**) Mean DVA activity (top) and mean locomotion speed (bottom). Shadings indicate SEM. Time is relative to $O_2$ downshift. n = 46 worms. (**C**) DVA neural activity changes in response to $O_2$ downshift using all data (top) or only during forward displacement, i.e. excluding reverse movement and pause phases (bottom). Quantifications of mean activity during 60s intervals pre- and post- $O_2$ downshift indicated by dashed lines in (**B**). Values from single recordings are shown in gray and population median ± interquartile range are shown in magenta. DVA activity significantly increases upon $O_2$ downshift, but not when animals are moving forward (****$p<0.0001$; ns, $p=0.12$, Wilcoxon matched-pairs signed rank test). (**D**) Traces show mean undulation amplitude and turning amplitude triggered to DVA activity peaks (t = 0 s) during forward displacement (see blue dots in (**A**) for example events). Shadings show SEM. Number of events and worms is indicated. Wilcoxon matched-pairs signed rank tests indicate significant changes (****$p \leq 0.0001$) between 1 s or 2 s intervals pre- and post-event as indicated.

The following figure supplement is available for figure 8:

*Figure 8 continued*

**Figure supplement 1.** Neural activity of interneuron DVA is associated with reversals and pausing phases.

Previous studies showed that *C. elegans* navigation in chemical and temperature gradients involves various navigational strategies such as modulation of reorientation rate, modulation of speed and finely controlled steering (*Iino and Yoshida, 2009; Luo et al., 2007*; *Luo et al., 2014a*; *Pierce-Shimomura et al., 1999*; *Schild and Glauser, 2013*). Since it was unknown which strategies contribute to $O_2$ chemotaxis, we performed a quantitative characterization thereof. Here and in the subsequent section, we focused on the low oxygen avoidance condition (16% $O_2$–4% $O_2$). Finely controlled steering during navigation of chemical gradients was previously described as a chemotaxis strategy termed weathervaning: animals introduce a curving bias towards preferred conditions depending on their heading direction with respect to the gradient direction (bearing) (*Iino and Yoshida, 2009*). Heading straight towards 16% or 4% $O_2$ was defined as 0° or 180° bearing respectively while heading perpendicular to the gradient was defined as 90° bearing (*Figure 9C*). We then calculated the animals' curving bias, which is the average change in the bearing trajectory $\Delta B/\Delta x$ as a function of bearing (*Figure 9C–D*). In comparison to control experiments lacking a gradient, animals exhibited a curving bias towards preferred $O_2$ concentrations, strongest at bearing orientations around 90° (*Figure 9D*). Next, we investigated the modulation of reorientation rate and found that reversal rates were suppressed or increased at low or high bearing angles, respectively (*Figure 9E*). We also observed a gradual reduction of locomotion speed as a function of increased bearing angles (*Figure 9F*). Finally, we detected individual shallow turning events based on finding local peaks of turning mode signal of the mid-body during forward locomotion, excluding omega turns. On average, turning increased gradually as a function of bearing angles during $O_2$ chemotaxis but not under control conditions (*Figure 9G*).

In summary, we found that wild type N2 animals when fasted for 1–1.5 hr perform $O_2$ chemotaxis to approach intermediate $O_2$ concentrations centered around 16.4%. The low $O_2$ avoidance mechanisms employ multiple navigational strategies: when heading along directions around perpendicular to the gradient animals steer towards preferred $O_2$ concentrations using weathervaning, and when animals are heading away from preferred $O_2$ concentrations (high bearing angles) the random biased walk mechanism of chemotaxis is prevalent as indicated by an up-regulation of reversal rates. In addition, animals exhibit enhanced ARS-like behavior indicated by a gradual reduction of locomotion speed and up-regulation of turning dependent on their heading direction.

## The AVK/FLP-1 and DVA/NLP-12 systems are required for spatial navigation in $O_2$ gradients

Based on our data, we hypothesized a role of the AVK/FLP-1 and DVA/NLP-12 systems for turning maneuvers during $O_2$ chemotaxis. *flp-1;nlp-12* double mutants showed a defect in their ability to distribute in response to the $O_2$ gradient (*Figure 10A*). We subtracted the fractional animal distributions during controls from the distributions during gradient application in order to calculate $O_2$ chemotaxis indices for low $O_2$ concentration ranges. This revealed a compromised ability of *flp-1; nlp-12* double mutants to avoid low $O_2$ concentrations (*Figure 10B*). A similar defect was observed in *nlp-12* and *flp-1* single mutants but not in AVK- animals (*Figure 10—figure supplement 1A–D*). However, *flp-1* single mutants as well as AVK- animals showed a stronger tendency to accumulate in the center of the assay arena during control experiments (*Figure 10—figure supplement 1B,C*). This effect appeared similar to crowding behavior animals perform under conditions of high population density and food scarcity (unpublished observation). Since this effect could potentially mask $O_2$ chemotaxis behaviors and thus might impede the interpretability of results we focused our further in-depth analysis on *flp-1;nlp-12* double mutants (*Figure 10*) and *nlp-12* single mutants (*Figure 10— figure supplement 1E–H*). *nlp-12* mutants as well as *flp-1;nlp-12* double mutants showed reduced weathervaning $O_2$ chemotaxis (*Figure 10C*; *Figure 10—figure supplement 1E*) but were able to modulate reversal rates (*Figure 10D*; *Figure 10—figure supplement 1F*) despite lower reversal rates in *nlp-12* single mutants (*Figure 10—figure supplement 1F*). *flp-1;nlp-12* double mutants did not modulate locomotion speed and turning mode as a function of bearing (*Figure 10E,F*); both

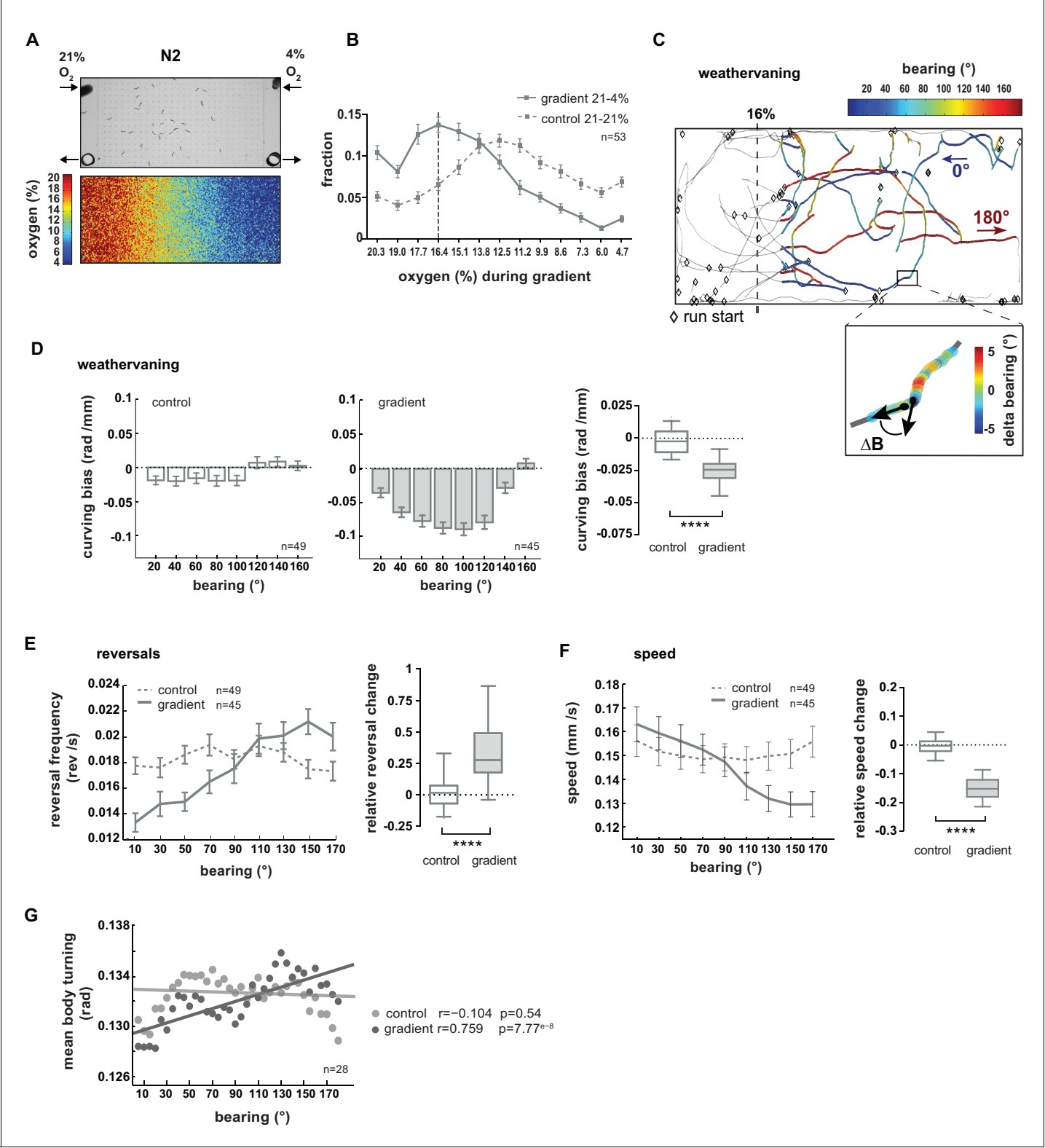

**Figure 9.** $O_2$ chemotaxis involves regulation of turning, reversals and locomotion speed. (**A**) $O_2$ chemotaxis assay. Top: video frame capture showing worms in $O_2$ gradient arena. Arrows indicate gas inlets and outlets. Bottom: Measured $O_2$ gradient in the device. (**B**) Distributions (trial-means ± SEM) of wild type N2 worms along the length of the arena. Data binning on the x-axis is 1.3% $O_2$. For gradient (solid line) or control (dashed line) assays either 21% and 4%, or 21% and 21% oxygen were applied at the inlets. Dashed line at 16.4% marks the bin center of the average peak accumulation during the gradient. n = 53 assays, 30-40 animals/assay. (**C, D**) Analysis of weathervaning chemotaxis. (**C**) Example trajectories of wild type N2 worms in

*Figure 9 continued on next page*

*Figure 9 continued*

the oxygen gradient. Color code indicates heading orientation (bearing) with respect to the $O_2$ gradient: bearing towards or away from 16% $O_2$ is defined as 0° or 180°, respectively. Gray trajectory sections were excluded from the analysis. Inset: example trajectory with color-code indicating the change of bearing ΔB. (**D**) Trial-mean curving bias ΔB per displacement (± SEM) under control (left) and gradient (middle) conditions as a function of bearing (20° binning). Right: quantification of curving bias averaged across a bearing range of 40°–150°. (**E**) Left: Trial-mean reversal frequency (± SEM) under control (dashed line) and gradient (solid line) conditions as a function of bearing (20° binning). Right: quantification of difference in mean reversal frequency below versus above 90° bearing. (**F**) Left: Trial-mean forward centroid speed (± SEM) under control (dashed line) and gradient (solid line) conditions as a function of bearing (20° binning). Right: quantification of difference in mean forward crawling speed below versus above 90° bearing. (**G**) Trial-mean mid-body (segment angle #11) signal amplitude of individual turning events under control (light gray) and gradient (dark gray) conditions as a function of bearing (5° binning). The linear fit and the correlation coefficients (r) are indicated; corresponding p-values indicate significance of correlation. The indicated sample sizes are n = number of experiments. All boxplots display trial-median, interquartile range and 5–95 percentile whiskers. Quantifications of control and gradient experiments are significantly different by Mann-Whitney test (****p<0.0001).

parameters were modulated in *nlp-12* single mutants despite low and high basal rates in speed and body turning signal, respectively (*Figure 10—figure supplement 1G,H*).

Taken together, these results show that when both FLP-1 and NLP-12 signaling is abolished, animals exhibit specific defects in navigating $O_2$ gradients: both weathervaning at intermediate bearing angles and up-regulation of ARS-like behaviors, i.e. slowing and increased turning, at high bearing angles are compromised; however, modulation of reversal rates as a function of bearing angle is unaffected.

## Discussion

A detailed description of motor patterns underlying animal locomotion during LDT and ARS contributes toward understanding the neural operations that execute the animals' strategy to navigate in the environment. Here we report on a posture control system in *C. elegans* that coordinates two elementary motor modes to drive efficient dispersal or local search behaviors, respectively. This coordination is required for foraging and goal-directed chemotaxis.

We show that a sudden decrease in environmental $O_2$, detected by BAG $O_2$ sensory neurons, evokes an array of behavioral changes in food-deprived animals (*Figures 1A–E*; *Figure 1—figure supplement 1*). Unlike previously described brief escape reflexes (*Goodman, 2006*; *Hilliard et al., 2002*), these behaviors are sustained for several minutes and can be described as a switch from LDT to $O_2$-induced ARS, in accordance to previous literature studying similar behavior strategies after removal from food (*Calhoun et al., 2015*; *Gray et al., 2005*; *Hills et al., 2004*; *Peliti et al., 2013*; *Tsalik and Hobert, 2003*; *Wakabayashi et al., 2004*). The reduction in ambient oxygen could indicate the proximity of a food source and therefore evoke similar ARS behavior as observed after removal from food. In both paradigms the search strategy is thus likely built on the vicinity of a potential food source. However, whether the same interneuron circuits are controlling the two ARS behaviors has not been focus of this study and remains to be investigated. Yet, we know that NLP-12 neuropeptides play a role in both $O_2$-induced ARS (this study) and ARS after removal from food (*Bhattacharya et al., 2014*) indicating some potential overlap between both circuits.

The bending of each body segment of *C. elegans* is tightly coupled to the activity of the underlying body wall muscles. Therefore, posture time series are an indirect readout of muscle activity patterns (*Butler et al., 2014*). Here we show that strikingly different movement strategies of *C. elegans* can be described by projecting posture time series to a common basis of EWs (*Figure 2A*). We reconstruct two motor modes from these projections: reconstructing posture time series from EW1-2 reveals the regular undulation mode of the animal consisting of sinusoid-like postures (*Figure 2C, D*). The residual movement (turning mode) is a set of motor patterns with distinct characteristics (*Figure 2—figure supplement 1*), capturing the animals' curving and turning movements. Strikingly, complex postural time series can be additively composed out of these two simpler modes (*Figure 2D*, *Video 2*): The eigenworm approach decomposes the biphasic temporal profile of animal posture observed upon sensory stimulation into two reciprocally regulated, simpler, monophasic components (*Figure 4A*). This indicates that the undulation and turning modes may explicitly map to neural control mechanisms. We speculate that summation of distinct neural activity patterns might be an elegant solution for a neural system to generate complex movements. In this view the turning

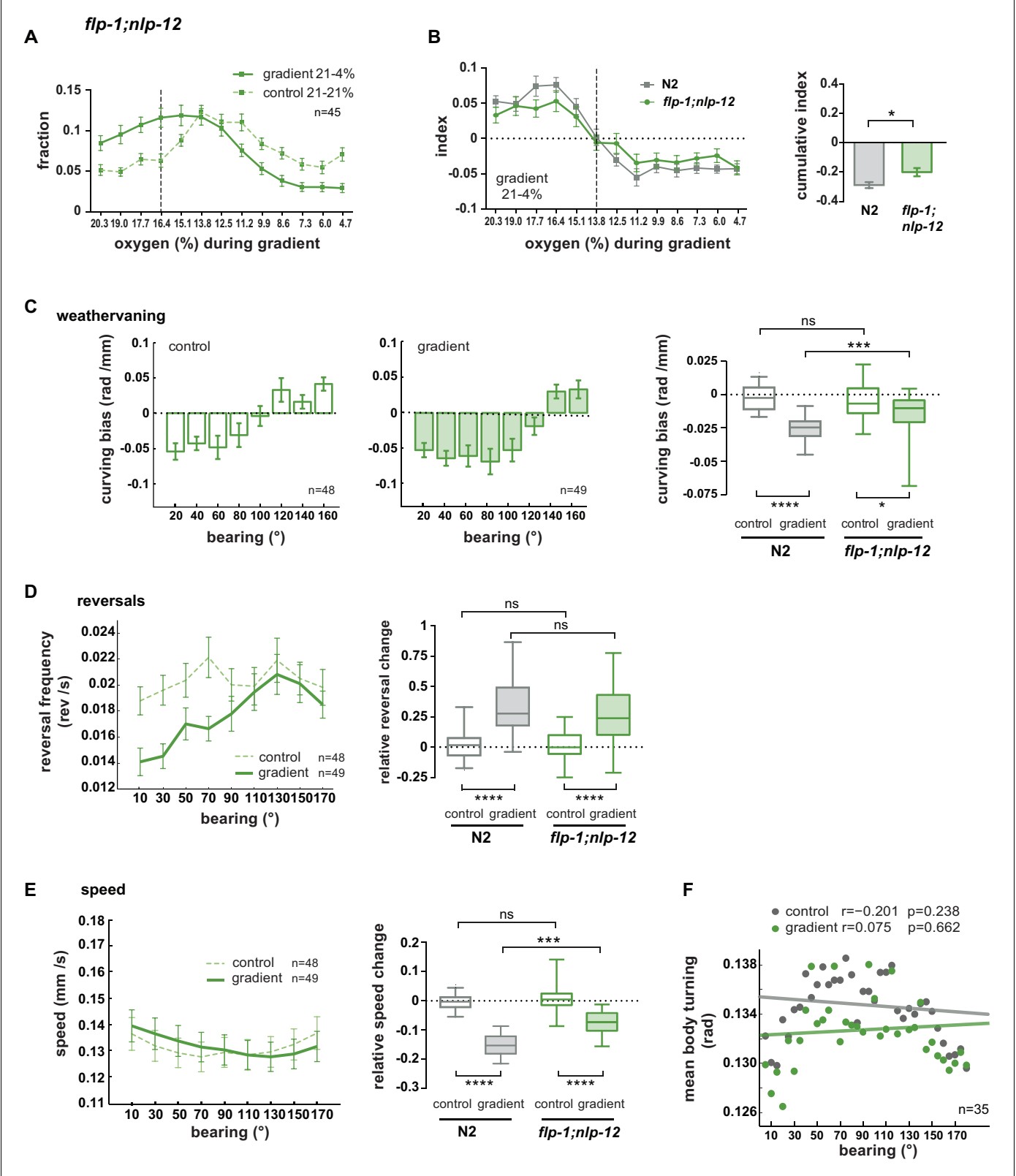

**Figure 10.** FLP1 and NLP-12 neuropeptides are implicated in the regulation of turning and locomotion speed during $O_2$ chemotaxis. (**A**) Distributions (trial-means ± SEM) of *flp-1;nlp-12* mutants along the length of the arena. Data binning on the x-axis is 1.3% $O_2$. For gradient (solid line) or control (dashed line) assays either 21% and 4%, or 21% and 21% oxygen were applied at the inlets. Dashed line at 16.4% marks the bin center of the average

*Figure 10 continued on next page*

Hums *et al.* eLife 2016;5:e14116. DOI: 10.7554/eLife.14116

*Figure 10 continued*

peak accumulation of wild type animals during the gradient. n = 45 assays, 30-40 animals/assay. (B) Left: Indices (trial-means ± SEM) of worm distributions along the length of the arena of wild type N2 (gray) and *flp-1;nlp-12* mutant (green) animals, calculated by subtracting in every bin the fraction of the paired control experiment from the fraction of the respective gradient experiment. Dashed line at 13.8% $O_2$ marks equal average accumulation during control and gradient conditions. Right: Cumulative indices (summed indices of all bins below 13.8%) of mutant and wild type N2 are compared by one-way-ANOVA with Dunnett's correction (*p=0.035, ***p=0.0003). All strains shown in *Figure 10—figure supplement 1D* were included in the statistical analysis. (C) Mean curving bias (± SEM) of *flp-1;nlp-12* mutants under control (left) and gradient (middle) conditions as a function of bearing (20° binning). Right: quantification of curving bias averaged across a bearing range of 40°–150°. (D) Left: Trial-mean reversal frequency (± SEM) of *flp-1;nlp-12* mutants under control (dashed line) and gradient (solid line) conditions as a function of bearing (20° binning). Right: quantification of difference in mean reversal frequency below versus above 90° bearing. (E) Left: Trial-mean forward centroid speed (± SEM) of *flp-1;nlp-12* mutants under control (dashed line) and gradient (solid line) conditions as a function of bearing (20° binning). Right: quantification of difference in mean forward crawling speed below versus above 90° bearing. (F) Trial-mean mid-body (segment angle #11) signal of individual turning events of *flp-1;nlp-12* mutants under control (light gray) and gradient (dark gray) conditions as a function of bearing (5° binning). The linear fit and the correlation coefficients (r) are indicated; corresponding p-values indicate significance of correlation. The indicated sample sizes are n = number of experiments. All boxplots display trial-median, interquartile range and 5–95 percentile whiskers. Quantifications of control and gradient experiments per strain are compared by Mann-Whitney test. Comparisons between *flp-1;nlp-12* mutant and N2 are performed with Kruskal-Wallis test with Dunn's correction (****p≤0.0001, ***0.0001<p≤0.001, **0.001<p≤0.01, *0.01<p≤0.05, ns p>0.05). Results of wild type N2 worms (shown in gray) used for comparison to mutants rely on the same data shown in *Figure 9*.

The following figure supplement is available for figure 10:

**Figure supplement 1.** Regulation of turning during $O_2$ chemotaxis depends on NLP-12 neuropeptides.

and undulation modes could have specific neural correlates at the level of interneurons and/or motor neurons, which are summed at the respective downstream postsynaptic targets. This hypothesis bears analogy to the proposal of muscle synergies producing coordinated movements in vertebrates (*Tresch and Jarc, 2009*). Further imaging studies simultaneously recording the activity of muscles, motor neurons and interneurons are required to test this hypothesis.

We describe three basic parameters of *C. elegans* locomotion: frequency and amplitude of the undulation mode contributing to the animals' locomotion speed and the turning amplitude contributing to changes in the direction of locomotion. All three parameters are in a lawful relationship: high undulation frequency is associated with high undulation amplitude, and both parameters are counter-regulated with turning amplitude (*Figure 3C,F*). This finding demonstrates an important tradeoff between flexibility and efficiency of movement: explorative behaviors with frequent directional changes require flexible postures and slower locomotion while fast undulation for efficient displacement requires movements coherent across all body parts. Therefore, undulation and turning motor modes are counter-regulated; however, they are not mutually exclusive as their contributions to movement are graded (*Figure 3–4*). Such counter-regulation might not be surprising, yet it has not been thoroughly described how animals in general coordinate underlying motor patterns for their purposes. Furthermore, counter-examples with co-regulation of those motor patterns exist in animal locomotion: e.g. hares sharply turn corners during fast escape runs, which suggests the need of neural coordination mechanisms of motor patterns.

Based on our findings we propose that LDT and ARS are not discrete alternative behavioral states, between which the animals switch, but rather represent extremes of a behavioral spectrum reflecting the animals' ability to flexibly adapt locomotion to their instantaneous needs. This contrasts with descriptions of *C. elegans'* roaming and dwelling behaviors in the presence of food, which have been characterized as discrete alternative behavioral states maintained for seconds to minutes, which is regulated by the neuromodulators serotonin (5-HT) and PDF neuropeptides (*Ben Arous et al., 2009*; *Flavell et al., 2013*). However, a different study has reported behavioral intermediates between roaming and dwelling (*Gallagher et al., 2013*). Nevertheless, roaming and dwelling characterized by ARS-related behavioral features on food might be different from $O_2$-induced ARS and LDT of food-deprived animals; it remains to be tested whether neural mechanisms, such as 5-HT and PDF signaling in roaming-dwelling decisions or FLP-1 and NLP-12 signaling in $O_2$-induced ARS, are shared between or exclusive for the two paradigms.

Our data show that undulation and turning modes can be dissected at the genetic and cellular level. We identified peptidergic neurons and neuropeptide genes, the genetic manipulations of

which strongly affect the animals' undulation and turning motions. Our data support the following model: FLP-1 neuropeptides released from AVK interneurons promote a shallow posture by constitutively restricting turning and undulation amplitude. AVK neuronal activity promotes undulation speed and suppresses turning (*Figure 5*, *7* and *Figure 5—figure supplement 2*). Unlike other speed encoding interneurons in *C. elegans* that selectively encode the speed of forward or reverse movement (*Kato et al., 2015*; *Li et al., 2014*) AVK encodes speed during the execution of both gaits. The decreased activity after stimulation is consistent with an increase of turning amplitude during ARS. Changes of AVK activity after sensory stimulation are evident only in unrestrained animals. Therefore, AVK might function as part of a locomotor feedback system to monitor speed changes and to inversely couple turning motions to these changes. In addition, it is likely that AVK neurons receive input from their presynaptic partners, the RIG interneurons (*White et al., 1986*), which are among the major postsynaptic partners of BAG sensory neurons and candidate transducers of $O_2$ stimuli (*Kato et al., 2015*), which could be a pathway for sensory-evoked postural control.

The DVA interneuron and NLP-12 neuropeptides secreted by it have differential effects on both motor patterns: our data support a model in which DVA activity/NLP-12 during forward locomotion promote undulation movements and locomotion speed while counteracting turning movements (*Figures 5*, *8* and *Figure 5—figure supplement 2*). In freely moving animals, we discovered various behavioral correlates of DVA neural activity, measured by $Ca^{2+}$-imaging, including pausing, reversing as well as transient reciprocal changes in undulation and turning amplitude (*Figures 8* and *Figure 8—figure supplement 1*). In our recordings DVA activity does not seem to encode sensory stimulus as it corresponds best to the execution of the various behaviors, the frequency of which changes upon stimulation. So far no candidate sensory to motor pathways from BAG to DVA have been characterized. DVA has been described as a proprioceptive neuron activated by body flexure in semi-restrained worms (*Li et al., 2006*). In our data obtained in freely moving worms we did not find a relationship between total body flexure and DVA activity. It therefore remains to be shown to what extent proprioception contributes to DVA activity in unrestrained animals.

When both characterized regulatory systems are compromised simultaneously, either in *flp-1;nlp-12* double mutants or in AVK-;DVA- animals, most locomotion parameters during LDT are either unaffected or only marginally affected (*Figure 5* and *Figure 5—figure supplement 2*). These results suggest that both signaling systems act in parallel but antagonistically. Yet, the ability of these animals to control locomotion speed, undulation amplitude and turning amplitude in response to stimulation is drastically affected (*Figure 5—figure supplement 2*, *Figure 5E–G*, *Figure 5—figure supplement 3*). These data indicate the importance of a counter-regulatory system including AVK/FLP-1 and DVA/NLP-12 for controlling undulation and turning motions in response to environmental changes.

We hypothesize that NLP-12 and FLP-1 exert their effects on undulation and turning motions by modulating neurotransmission at neuromuscular junctions: NLP-12 peptides have been shown to enhance synaptic transmission at cholinergic motorneuron synapses (*Bhattacharya et al., 2014*; *Hu et al., 2011*; Hu et al., *2015*). A similar effect on GABAergic motor neuron synapses has been suggested for FLP-1 peptides (*Stawicki et al., 2013*).

We show here that 1 hr food-deprived wild type N2 animals, unlike well-fed N2 animals (*Gray et al., 2004*), navigate in a linear $O_2$ gradient towards a concentration range centered around 16% (*Figure 9A,B*). This is a laboratory condition associated with abundant bacterial food (*Gray et al., 2004*), which suggests that fasted animals utilize information about ambient $O_2$ concentrations in order to search for food. We show that *C. elegans* employs multiple strategies during $O_2$ chemotaxis, the choice of which depends on how the animals are oriented along the gradient: when their orientation is close to perpendicular to the gradient direction, the weathervaning strategy prevails (*Figure 9D*). When heading straight up or down the gradient, they employ a biased random walk strategy by down- or up-regulating reorientation rate, respectively (*Figure 9E*). Similar observations have been made in other chemotaxis paradigms in *C. elegans* (*Iino and Yoshida, 2009*). In addition, we found that heading away from preferred $O_2$ levels causes the animals to slow down and up-regulate turning movements, i.e. they engage in an ARS-like behavior (*Figure 9F,G*). *flp-1;nlp-12* double mutants exhibit specific defects in weathervaning, speed modulation and up-regulation of turning movements, while the random biased walk mechanism remains intact (*Figure 10C–F*). The overall consequence is a decrease in chemotaxis performance (*Figure 10A,B*). These data indicate that the fine-tuned control of undulation and turning motions is essential for normal navigation of

spatial gradients; nevertheless, the biased random walk strategy can partially compensate for these deficits. In bacteria the biased random walk strategy is the only reported mechanism supporting the organism to reach its goal (*Berg and Brown, 1972*). It will be interesting to find out whether similar counter-regulation of motor patterns underlies the chemotactic strategies of other animals, e.g. *Drosophila* larvae. These move via whole-body propulsions and are also capable of controlling run directions (= weathervaning) besides regulating timing and direction of whole-body turns in response to odorant gradients (*Gomez-Marin et al., 2011*; *Gomez-Marin and Louis, 2014*).

In conclusion, we propose that combination of elementary motor patterns enables animals to flexibly adjust their locomotion strategy. The undulation mode dominates LDT while turning movements shape $O_2$-induced ARS. However, these behaviors lie on a continuum. Neuromodulatory peptidergic circuits gradually adjust the underlying movement parameters while enforcing lawful reciprocal relationships between them. In this way, movement can be finely controlled according to the animals' instantaneous needs, allowing for a rapid sensory-evoked shift from LDT to $O_2$-induced ARS as well as for subtle movement changes during navigation. Our approach is thus complementary with others that describe different behaviors as discrete and maintained behavioral states between which animals switch (*Ben Arous et al., 2009*; *Brown et al., 2012*; *Dankert et al., 2009*; *Kain et al., 2013*; *Wiltschko et al., 2015*). The coordination of motor patterns for foraging accounts for a tradeoff that animals have to make in order to effectively achieve the goals of each strategy: fast undulation requires a coherent coordination of all body parts, while slow search movements require flexibility in gait and posture. In this work we provide experimental support for this concept obtained from *C. elegans* and speculate that analogous control schemes may govern locomotion in higher animals.

## Materials and methods

### Worm population behavioral assays

Behavioral studies of *C. elegans* populations were done as described previously (*Zimmer et al., 2009*) with some modifications for increased image resolution and improved stimulus delivery: For each assay ~25 adult animals grown on OP50 seeded food plates (1 day post L4 larval stage) were transferred (via manual picking) without food onto a plane food-free nematode growth medium (NGM) agar surface in a 14 cm petri dish (NGM assay plate). Animals were starved for one hour on the NGM assay plate prior to examination. A 36 mm x 36 mm area was cut out of Whatman filter paper soaked in 20 mM of repelling $CuCl_2$ to prevent animals from leaving the assay arena. A custom-made transparent plexiglass device with a flow arena of 39 mm x 39 mm x 0.7 mm was placed onto the assay arena and animals were exposed to a gas flow of 25 ml/ min containing 21% (v/ v) $O_2$ for six minutes, followed by a switch to 10% $O_2$ for six minutes (downshift shift assays) or followed by a temporal ramp from 21% to 4% $O_2$ lasting three minutes (step size 0.094%/s). All gas mixtures were balanced with $N_2$. Gases were mixed with a static mixing element connected to mass flow controllers (Vögtlin Instruments, Aesch, Switzerland) that were operated by custom written LabVIEW (National Instruments, Austin, TX) software. The temporal $O_2$ shifts and ramp were confirmed by measuring $O_2$ concentrations in the device with an $O_2$-sensitive fluorescent spot sensor (PreSens, Regensburg, Germany). We measured that $O_2$ shifts equilibrate the arena within 12 s. The $O_2$ ramps were found to be linear as expected. Recordings of freely behaving animals illuminated with 200 mm x 200 mm flat red LED lights were made at 10 fps on 4–5 megapixel CCD cameras (JAI, Copenhagen, Denmark) using Streampix software (Norpix, Montreal, Canada). The pixel resolution was 0.0129 mm/ pixel.

Movies were analyzed with a custom written image processing and tracking code in MATLAB (MathWorks, Natick, Massachusetts). It is build upon a previously reported script (*Ramot et al., 2008*; *Tsunozaki et al., 2008*), available at http://med.stanford.edu/wormsense/tracker/. Briefly, worms were detected by gray level thresholding. Worm trajectories were generated by connecting nearby centroid coordinates in adjacent frames and each trajectory coordinate was assigned with recorded binary images, centroid coordinates and shape parameters; the resulting data-structures are termed here worm tracks. Trajectories are terminated when worms collide with each other or with the boundaries of the arena. Each worm track therefore represents a fragment of each worm's complete behavior during the recording time. Worm tracks with a length of less than 200 (20 s duration) frames were discarded. Each worm is typically represented by multiple worm tracks of varying

length. This is important to keep in mind when displaying and quantifying the data (see below). The resulting trajectories were smoothed and used to calculate instantaneous translational speed of the worm's centroid, which is measured along the axis of progression. Periods of backward locomotion were detected based on changes in angular velocity and building on the fact that animals were moving most of the time in forward direction, which was confirmed also for all mutants used in this study. Reversal frequency was calculated in 10 s bins. Reversals were usually excluded (data set to NaN) from the population behavioral assay analyses to obtain forward locomotion only. Time periods during which the animals were within 80pixels (= 1.0 mm) distance to the Whatman paper were also excluded. Deep, so-called omega turns were detected based on characteristic changes in object eccentricity and angular speed, and their frequency was calculated in 10 s bins.

Track curvature was determined from smoothed centroid trajectories for every frame as absolute change of heading angle between adjacent frames. Therefore the heading angle per frame $i$ was extracted as the four-quadrant inverse tangent of the x/y-coordinates resulting from the differences between the centroid x/y-positions in frame $i$-5 and frame $i$+5 (1 s intervals). The result (change of heading angle per frame) was divided by the mean crossed distance per frame during the same 1 s interval. This yielded change of heading angle over distance. Data during which animals were barely moving (speed < 0.03 mm/s) had to be excluded as centroid 'wobbling' due to tracking issues at very low speeds caused falsely high curvature levels. Artificially high values (>31 rad/mm = 99.5 percentile) were removed. Data during reverse movement were excluded.

Displacement was calculated in 30 s bins. Distance of the centroid positions between start and end point for every bin were determined for each worm track. For displacement, movement in reverse direction was not excluded.

## Skeletonization and segment angle analyses

After thresholding, the binary worm images were processed (by dilation, erosion, edge-smoothing, bridging unconnected points, hole-filling) before they were eventually skeletonized (including branch trimming for optimization) to obtain one-dimensional splines tracing the midline of the worms from head to posterior end. The pointy tail tips were omitted due to thresholding issues. The extracted splines were smoothed and divided into 25 equally spaced body segments. Artificially short skeletons were excluded and the data of these frames set to NaN. Images of worms forming coil-like postures could not be skeletonized. This typically occurred during deep bended omega turns when the head touches the tail. However, the most highly curved posture typically happened slightly before so that body posture, undulation mode and turning mode amplitude maxima during omega turns were still included in most of the data. Therefore, time points during those turning events before and after the moments of head-body touching could be skeletonized. Head positions were determined based on direction of movement and taking into account when the animals moved backward. Then skeletons were reordered accordingly. Head-tail flips were further prevented due to the fact that the head position could only change by limited distance in-between adjacent frames (0.1 s): both skeleton end points were compared to the head position of the previous frame and the end point with smaller distance was usually (except for identified deep omega turns, when head and tail position were very close) taken as the new head position. Manually proof checking confirmed the reliability of this method. We calculated 24 inter-segment angles between the adjacent segments from head (segment angle #1) to tail (segment angle #24). Segment angle time series were added to the worm tracks structures. They were termed body posture. Previous studies had used at least 48 (up to 100) segments (*Brown et al., 2012*; *Stephens et al., 2008*; *Yemini et al., 2013*). In this paper, we can recapitulate similar eigenworm shapes with 24 segment angles. The advantage of choosing (in comparison to these studies) lower resolution was being able to record from multiple animals simultaneously and to acquire data with the parallel worm tracker described above. This approach provides higher sample power required for trial averaging and statistics when assaying behavioral responses that naturally exhibit high single trial animal-to-animal variability.

## Eigenworm-based decomposition and analysis of undulation and turning mode

For calculation of eigenworms (except for *Figure 2A,B*), all wild type N2 segment angle time series (containing forward and backward locomotion) were concatenated. For *Figure 2A,B*, only wild type

segment angle time series that were spanning defined 60 s intervals directly pre- or post- $O_2$ shift, respectively, were concatenated. Then eigenworms (EW) (and eigenvalues for *Figure 2A,B*) were derived by standard principal components analysis (PCA) on the concatenated angle time series using MATLAB. Variance contribution per eigenworm (*Figure 2A,B*) was calculated as the relative fraction each eigenvalue contributed to the sum of all eigenvalues. The difference between the cumulative variance distributions of the two 60 s intervals was evaluated for significance by a random resampling approach: the same angle time series were shuffled and then randomly split into two groups with n numbers matching the original counts of time series pre- or post-shift, respectively. Then PCA was performed on the concatenated angle time series per group (as before) and the cumulative difference between the two cumulative variance contributions was calculated. These steps were iterated one million times during which an equal or higher value than the one obtained from the original non-shuffled groups never occurred. This procedure thus yielded an estimate of an upper bound of $10^{-6}$ for a p-value.

Segment angle time series of all individual worm tracks from all strains were projected onto these wild type-derived eigenworms. The time series mean of each angle was subtracted from each angle. Then, undulation mode or turning mode was generated by firstly calculating the cross-product of the 24 segment angles with a matrix made of eigenworms 1–2 or remaining eigenworms 3–24, respectively. This step yielded a description of the body posture in terms of 2 or 22 projection amplitudes along the respective eigenworms. Secondly, the cross product of the result with the respective eigenworm matrix was calculated and the previously subtracted mean was added. This step retrieved the description in terms of 24 segment angles.

$$undulation\,mode = [\text{EW1} : \text{EW2}] \times ([\text{EW1} : \text{EW2}]^{\text{T}} \times [\text{angles} - \text{mean(angles)}]) + \text{mean(angles)}$$

$$turning\,mode = [\text{EW3} : \text{EW24}] \times ([\text{EW3} : \text{EW24}]^{\text{T}} \times [\text{angles} - \text{mean(angles)}]) + \text{mean(angles)}$$

This procedure decomposed the body posture on a frame-by-frame basis into undulation mode and turning mode. As PCA is a linear decomposition method, the total body posture is the sum of every corresponding undulation and turning mode:

$$body\,posture = undulation\,mode + turning\,mode$$

Cross correlation functions in *Figure 2—figure supplement 1* were calculated from selected segment angles (as indicated) of all wild type N2 undulation or turning mode time series during forward locomotion. This method was robust with respect to the varying lengths (at least 20 s) of worm tracks.

Phase velocity (= undulation frequency) of undulation mode was derived by calculating the cross-product of the 24 segment angles time series (after subtracting the time series mean of each angle) with the eigenworms 1–2. The obtained two projection amplitudes $a_1$ and $a_2$ oscillate in quadrature (*Stephens et al., 2008*). They were gently smoothed and the phase angle for each frame was derived from the cross-product of vectors in the ($a_1$, $a_2$) space from adjacent frames; then the phase velocity was calculated from the obtained angles as cycles (360° angle) per time (cycles/second).

We calculated the amplitude of body posture, undulation and turning mode by summing the absolutes of all 24 segment angles per time point.

In order to account only for forward locomotion, periods of backward locomotion were excluded from population data.

Worm shapes in *Figure 2D* and in *Video 2* were reconstructed from total body posture, undulation mode and turning mode: synthetic worm silhouettes were calculated at each point in a time series as follows. From the 24 segment angles and a list of 25 fixed end-to-end segment lengths, the position of 26 2D points along the central line of the worm body from head to tail were calculated. Combining these 26 central line points with a fixed list of 26 cross sectional widths from head to tail, a filled 2D polygonal model of the worm consisting of 50 quadrilaterals and 2 triangles, one for the head-most segment and one for the tail-most segment, was created. The overall orientation for these reconstructed worms was determined at each time point by matching the centroid-to-nose-tip vector to a corresponding vector extracted from real movies. The filled 2D polygonal model was then rendered into an image using a standard triangle-filling pixel scan-line algorithm.

## Distributions, trial averaging, quantifications and statistics of behavior data

For averaging, means and standard errors of the mean (SEM) of behavioral time courses were determined on a frame-by-frame basis from all worm tracks of all experiments available in each frame, for centroid speed, amplitudes and track curvature. These averages were finally binned (by 10 frames = 1 s intervals) for display reasons. Averages of reversal and omega turn frequency or displacement rate (*Figure 1—figure supplement 1A,B* or *Figure 1C*) were determined by averaging worm track data derived from 10 s or 30 s bins, respectively.

Distributions of behavioral features were derived as follows. For relative distributions of turning amplitude during short time intervals (*Figure 4C*), all data from the indicated time interval were pooled per experiment and normalized histograms with 0.1 rad bin size were derived and then averaged over all experiments. For analyzing two behavioral features against each other, all worm tracks (binned by 5 frames = 0.5 s) within the respective indicated time interval of all experiments were pooled. For density maps displaying 2D distributions of undulation frequency against turning amplitude (*Figure 4D*), normalized 2D histograms were derived with the bin sizes of 0.002 cycles/s for undulation frequency and 0.025 rad for turning amplitude. Very high values of undulation frequency (>0.6 cycles/s) and amplitude (>4 rad) were not shown due to data sparseness. For density maps displaying 2D distributions of undulation amplitude against turning amplitude (*Figure 5—figure supplement 1*), normalized 2D histograms were derived with bin sizes of 0.05 rad. A boundary encompassing the 99% most abundant data of wild type N2 was calculated. The per cent of data of manipulated strains lying within this wild type N2 boundary was determined. For sorting one behavioral feature Y over another feature X (*Figure 3*), feature X was divided into bins of equal length (undulation frequency 0.015 cycles/s; turning amplitude 0.2 rad [against track curvature]; undulation amplitude 0.1 rad [against turning amplitude]). Medians and inter-quartile ranges of feature Y from all worm track data within each bin of X were calculated and plotted over the medians of bins from X. Linear correlation coefficients were calculated via a MATLAB standard function. Example trajectories in *Figure 3A* were chosen during the 10% $O_2$ phase. Positions and turning amplitude were binned by 5 frames (0.5 s) and the trajectories' start points were aligned.

For quantifications and statistics we chose to account for experiment-to-experiment variability. Therefore the population mean for each individual experiment was calculated from respective indicated time intervals. For changes after $O_2$ downshift or through the $O_2$ ramp, the means of two equally sized intervals were subtracted per experiment. Relative changes of locomotion speed were determined by normalizing the change over the level obtained during the interval pre-downshift for each experiment. Sample sizes were the number of experiments. All data of mutants were compared to the wild type N2 dataset, or between selected strains as indicated, by one-way-ANOVA with Sidak correction. For comparing neuropeptide mutants and cell ablated lines to wild type (*Figure 5*), all acquired data per strain were used, because the *flp-1* mutant displayed relatively increased behavioral variability between different sets of experiments. In order to get the best picture of its phenotype all data were taken into account. For analyzing neuropeptide rescues strains (*Figure 6*), only the subset of mutant and the subset of wild type experiments performed in parallel to the rescue strains were used. Only for the rescue of *flp-1;nlp-12* mutants (*Figure 6*), exclusive datasets of double rescue, mutant and wild type strains were acquired.

For comparing intervals pre- and post-shift per strain (*Figure 1—figure supplement 1*), paired t-tests were performed.

## Spatial $O_2$ gradient assays

For $O_2$ chemotaxis assays we used a previously reported $O_2$ gradient device made of polydimethylsiloxane (PDMS) (*Gray et al., 2004*). We chose different experimental conditions than in previous studies (*Chang and Bargmann, 2008*; *Chang et al., 2006*; *Gray et al., 2004*; *Zimmer et al., 2009*) in order to optimally match all conditions across behavioral experiments in this study. The important parameters were: low population density, 1 hr food deprivation and no hypoxia conditions in the device (21-4% instead of 21-0% gradients). Hence different preferred $O_2$ levels are reported here (16–17% in this study versus 7-10%). Animals were starved for 1 hr on an NGM agar plate without *E. coli* feeding bacteria before the aerotaxis assays. In each assay, 30–40 adult (1 day post L4 larval stage) animals were transferred onto a new NGM agar plate and the PDMS device was placed over

them. This device comprised an arena of 33 × 15 mm. Gas in- and outlets were included within 3 mm wide stretches on either side of the arena (*Figure 9A*). The gas was producing a linear gradient by diffusion. 21% or 4% (v/v) $O_2$, balanced with nitrogen, was delivered with a flow rate of 0.75 ml/min from gas-tight syringes using a syringe pump (PHD2000, Harvard Apparatus). Each gradient experiment was accompanied by a preceding control experiment performed with application of 21% $O_2$ on both sides. After performing control experiments for 30 min, gradients were established and animals were recorded for another time course of 30 min. The $O_2$ gradient establishment was confirmed in separate measurements with the VisiSens $O_2$ imaging system (PreSens) by ratiometric fluorescence imaging of an $O_2$-sensitive foil covered with an agarose film and the gradient device. These experiments revealed that a nearly linear gradient is established within 10 min. Recordings of freely behaving animals illuminated with flat red LED lights were made at 3, 10 or 15 fps on a 4 megapixel CCD camera (Jai) at a pixel resolution of 0.0276 or 0.0155 mm/ pixel using Streampix software (Norpix).

## $O_2$ chemotaxis analysis

For worm population profiles of gradient and control experiments, one video frame at 30 min of the recording was evaluated. Animals within 13 equally spaced 2-mm bins of the assay arena were manually counted and the fraction of animals per bin was calculated for every experiment. An index was calculated by subtracting per bin the fraction of the respective pre-run control experiment from the fraction of each gradient experiment. Means and SEM were calculated from all experiment replicates. Sample sizes were the number of experiments with available paired controls. For quantification, cumulative indices were determined per experiment by summing the index from all bins below, respectively, the bin that displayed a mean index of zero for wild type animals (13.8% $O_2$ at its center). Cumulative indices of mutant strains were compared to wild type indices by one-way-ANOVA with Dunnett's correction in separate tests.

For further analyses, movies were analyzed as described above for worm population behavioral assays. Bearing (B, measured in degree°) was quantified as the instantaneous orientation of the animals' smoothed centroid heading relative to the optimal $O_2$ concentration isoline. We defined the metric range from 0° (orientation exactly towards the isoline) to 180° (orientation away from the isoline). Left or right orientation was neglected, as the gradient in the arena is one-dimensional. We excluded sections of the trajectories that were within 20 pixels (=0.55 mm) distance to the arena edges. Our weathervaning analysis was restricted to the area of the gradient with concentrations lower than 16% and 2 min after $O_2$ flow onset. Curving bias was calculated as the instantaneous change of bearing over distance travelled dB/dX (rad/mm) for each frame. The analysis was restricted to continuous forward run periods: Omega turns, reversals, forward runs of very short duration (<3 s) or forward runs with a total displacement of less than 1 mm were excluded. A negative curving bias indicates a change of orientation towards the optimum isoline. Curving bias, reversal frequency and speed of forward runs relative to bearing were obtained by calculating histograms using bins of 20° bearing. For quantification and statistical comparisons curving biases were averaged across a bearing range of 40°–150°. Relative reversal and speed change was calculated as the percent change of reversal frequency or speed, respectively, for animals with a bearing >90° relative to the reversal frequency or speed of all animals with bearing <90°. These parameters were calculated independently for each experiment. Kruskal-Wallis test with Dunn's correction was used for comparing mutant strains against wild type N2 and Mann-Whitney test for comparing gradient vs. control for each strain. Sample size equals number of experiments (successfully worm-tracked control and gradient runs counted spearately). For calculating mean body turning with respect to the animals' bearing we used the high pixel resolution datasets (hence the lower n numbers in Figure 9G, 10F), skeletonized worms and performed eigenworm analysis as above. Peak values of the mid-body turning mode (segment angle #11) only during forward movement and excluding omega turns were detected with a peak-finding algorithm and averaged over bins of 5° for corresponding bearing values. Standard MatLab functions were used for the linear fit and calculation of linear correlation coefficients. *p* values are calculated based on a Student's t distribution using the *corr* function in MatLab.

## Simultaneous imaging of neuronal calcium and behavior

Calcium imaging recordings were made using an automatic re-centering system previously described (*Faumont et al., 2011*). Adult (1 day post L4 larval stage) worms expressing both mCherry and GCaMP5K (or GFP) in the neuron of interest were placed on food-free nematode growth medium (NGM) agarose pads and sealed in a custom built airtight chamber with inlet and outlet connectors for gas delivery. Animals were starved for 1 hr prior the imaging experiment on a food-free normal NGM plate. 21% (v/ v) $O_2$, balanced with nitrogen, was applied for 4 min with a gas flow of 50 ml/ min, followed by a switch to 10% $O_2$ for 4 min. Gases were mixed with a static mixing element connected to mass flow controllers (Vögtlin Instruments). To avoid out of focus movement it was critical to carefully prepare plane agarose pads. For this, freshly made NGM agarose mix was melted and poured into a ring 2.45 mm thick and 50 mm in diameter and enclosed with glass on both sides to harden. Another ring of 38 mm diameter was pressed onto the agarose in order to make an indentation into which 20 mM copper chloride was pipetted to restrict the worm to the center of the pad. The pad was then placed inside the chamber, which was sealed shut and covered with a glass slide (0.55 mm thickness) 0.7 mm from the agarose surface. The chamber was then placed onto a motorized stage with associated controller (MS-2000-PhotoTrack, Applied Scientific Instrumentation). Images were acquired on an inverted compound microscope (Zeiss Axio Observer.Z1) using two Charge-Coupled Device cameras (Evolve 512, Photometrics). Dual wavelength excitation light (470 and 585 nm) was provided by a CoolLED pE-2 excitation system using an ET-EGFP/mCherry filter set (59022x, Chroma) and dichroic (59022bs, Chroma). A long-distance 63x objective (Zeiss LD Plan-Neofluar 63x, 0.75 NA) was used to stream unbinned images at 33 ms exposure time (30 Hz) with Visiview software (Visitron Systems GmbH, Germany). A dichroic mirror (620 spxr, Chroma) directed high wavelength mCherry emission to a four-quadrant photomultiplier tube (Hamamatsu) for recentering. The remaining emission was split by a DualCam DC2 cube (565 lpxr, Photometrics) to each of the two CCD cameras, one for mCherry emission (641/75 nm, Brightline) and one for GCaMP emission (520/35 nm, Brightline). mCherry emission was further reduced to 50% by a neutral density filter to prevent signal saturation. Simultaneous behavior recordings under infrared LED illumination (780 nm) were made using an IR-sensitive CCD camera (Manta Prosilica GigE, Applied Vision Technologies) at 4x magnification with 100 ms exposure time (10 Hz) and StreamPix software (Norpix). The pixel resolution was 1.6 mm/ px.

## Image processing of freely moving calcium imaging data

We extracted fluorescence intensity values for GCaMP and mCherry with a custom-made MATLAB tracking script. We determined a threshold intensity value for the mCherry channel-derived images capturing the neuron of interest during all recording frames and then measured the sum of pixels from a connected region of at least 50-pixel size above threshold for each frame. For the GCaMP channel-derived images we measured the sum of pixels from the region matching the mCherry thresholded area. Both datasets were background-corrected by subtracting the average background pixel value (obtained from the respective first image frame capturing unspecific tissue signal) multiplied by the number of pixels thresholded in that frame. The ratio GCaMP5K/ mCherry was calculated per frame to correct for artificial GCaMP fluorescence changes due to motion artifacts or out-of-focus movements of the cell body. Very strong out-of-focus movements were automatically excluded, as there was no connected region >50 pixels above threshold in these frames. In that case data were set to NaN. We further corrected for strong out-of-focus movements, not fully adjusted through the ratio calculation, by determining strong drops in mCherry fluorescence and occasionally unreasonably large drops in GCaMP fluorescence values. Normalized ratio dR/R (%) was calculated as the difference of R extracted from each frame to the mean R of the total recording and divided by mean R [(R-mean(R))/mean(R)]. Further signal artifacts were determined and excluded by determining extremely high ratio values (absolute and relative to a finer local mean) and drops (relative) of the normalized ratio, which had been carefully evaluated for each cell. All excluded data were set to NaN. GFP control data were extracted and processed exactly the same way as GCaMP data.

## Segmentation, skeletonization and extraction of segment angles from infrared recordings

Worm skeleton extraction from infrared videos was performed as described (*Yemini et al., 2013*), save for a few minor modifications. The source code is available at https://github.com/openworm/ SegWorm. Videos were down-sampled to 520 × 519 pixels. The stereotypy of the images permitted choosing a fixed manual threshold (an 8-bit grayscale value of 127) for every recording to separate worm from background. The thresholded worm was dilated by 3 pixels so as to smooth any imperfections, and then eroded by 8 pixels to achieve an accurate representation of the worm. The original algorithm was too stringent in rejecting poorly segmented worm shapes. Therefore, all thresholds for worm-shape rejection were relaxed by 80% without compromising the effectiveness of this step. Head/tail classification was a matter of determining which end of the worm was more central. Therefore, within video chunks of contiguous worm segmentations, individual skeletons were oriented relative to each other as previously described, then head and tail were determined by taking the mean distance of both worm ends from the video center. Thereafter, all steps to skeletonize the worm are as formerly described, save for a reduction in skeleton size to 26 worm points.

## Analyses of simultaneous calcium imaging and behavioral data

Precise positions of the motorized tracking stage and thus the tracked neural target were recorded for every frame (30 fps) via the VisiView software and extracted using a MetaMorph (Universal Imaging) custom-made script. Obtained stage positions were further analyzed with MATLAB-based custom-made processing scripts. Locomotion speed was calculated with a step-size of 30 frames (=1 s), i.e. speed of frame $i$ was the distance between positions of frame $i$-15 and frame $i$+15 divided by the passed time. Artificially high values (>0.6 mm/s), when the tracking of the cell was lost, were excluded. Angular velocity was calculated with a step-size of 5 frames after extracting heading angles from changes in x/y-coordinates during 30-frame bins of smoothed (30 frames) trajectories. Then, periods of backward locomotion were detected based on characteristic changes of angular velocity, the speed derivative and speed, and by considering a maximum length of reversals per strain.

The splines encoded by the extracted 26 worm skeleton points were smoothed per frame and 24 inter-segment angles were calculated for each frame. When no or artificial (too long or short due to image segmentation issues) skeletons were retrieved, segment angles were set to NaN. Small gaps (<0.5 s) in segment angle time series were filled by cubic interpolation and angles were smoothed over time (15 frames = 1.5 s). In order to match calcium imaging and segment angle data, the segment angle time series were cubically interpolated from their original acquisition rate of 10 Hz to the 30 Hz-frame rate of the calcium imaging recordings.

Single traces of normalized ratio or amplitude were smoothed by 15 frames (0.5 s) prior trial-averaging. Then means and SEMs from time courses of normalized ratio dR/R or behavior parameters were calculated from all recordings and averages were binned (by 30 frames = 1 s intervals) for display reasons. Fraction of animals pausing were calculated and binned into 5 s bins. When indicated, only data during certain locomotion phases e.g. forward movement were used to calculate averages by setting ratio or behavior data e.g. during reversals (and pause phases) to NaN. For statistics, means of normalized ratio or behavior parameters were calculated per worm from respective indicated time intervals. The means pre- and post- $O_2$ downshift were compared with paired tests (paired t-test or Wilcoxon matched-pairs signed rank test) as indicated to evaluate the changes. For comparing these $O_2$-evoked changes between different worm strains (GCaMP- vs. GFP-expressing worms) or conditions (freely moving vs. immobilized) the means of the two equally sized intervals (pre and post) were subtracted per worm and evaluated by one-way-ANOVA with Dunnett's correction or Mann Whitney test, as indicated. Single worm traces for illustration were smoothed by 5 frames and every $10^{th}$ point was plotted for display reasons.

Eigenworms of all pooled segment angle time series (containing forward and backward locomotion) of wild type GCaMP-expressing N2 worms from calcium-imaging experiments were concatenated (separately for AVK and DVA imaging lines). Then eigenworms (EW) were derived by standard principal components analysis (PCA) on the concatenated angles using MATLAB. The individual segment angle time series of all recorded worms were projected onto these wild type-derived (respective AVK- or DVA-GCaMP) eigenworms exactly the same way, as done for the population

behavioral assays to retrieve undulation and turning modes. Further the amplitude of each mode was calculated accordingly by summing the absolutes of all 24 segment angles per time point.

For further analyses all data time series were smoothed by usually 5 frames, or 15 frames when peak detection was involved. For analyzing the relation of the neural activity ratio to selected behavior features, data points per worm were binned by 15 frames (= 0.5 s) and then pooled from all recordings. When indicated only data within respective time intervals or data during specific locomotion phases were taken into account. The ratio data were then sorted over bins of equal length of the behavior feature (speed 0.01 mm/s, amplitude 0.2 rad). Medians and inter-quartile ranges of the ratio data within each bin were calculated and plotted over the medians of the behavior bins. Very high values of speed (>0.35 mm/s) and amplitude (>4 rad) were not shown due to data sparseness. Linear correlation coefficients were calculated per worm with a MATLAB standard function and compared between intervals, locomotion phases or between strains, using paired or unpaired t-test as indicated. Boxplots display median, interquartile range and 5–95 percentile whiskers . Before analyzing the relation of AVK ratio and locomotion speed, ratio data were shifted relative to speed data by a lag time of 1.67 s, which had been derived from the peak r value of a performed cross-correlation of the two signals.

For event-triggered averages, a peak-detection algorithm was used to determine DVA ratio peaks during forward moving phases (from identified 'moving phases' and excluding reversals). Further reversal start time points derived from the stage position were used. The respective traces of amplitude plus surrounding 6–10 s on either side were aligned to these events and means and SEMs per frame of all events were calculated and plotted. Wilcoxon matched-pairs signed rank tests compared means calculated per event of short intervals (length as indicated) pre- and post- event. The means of the pre- and post- reversal start intervals were subtracted per event and the changes were evaluated by Mann-Whitney test comparing different GFP-expressing worms to GCaMP-expressing worms.

Moving and pausing phases were classified as phases of at least 2 s duration above or below a speed threshold of 0.05 mm/s, tolerating gaps of maximum 1 s. Means of normalized dR /R during moving or pausing phases, from worms that were pausing at least 60 s of the recording time (with interruptions), were compared via Wilcoxon matched-pairs signed rank tests.

## Imaging of neural activity in microfluidic chip

Microfluidic two-layer PDMS devices were constructed as previously described (*Chronis et al., 2007*; *Zimmer et al., 2009*). The worm channel was connected to a reservoir containing S-Basal buffer. All components were connected with Tygon tubing (0.02 in ID, 0.06 in OD; Norton) or polyethylene tubing (0.066 in ID, 0.095 in OD; Intramedic) using 23G Luer-stub adapters (Intramedic). 21% (v/v) $O_2$, balanced with nitrogen, was applied with a gas flow of 50 ml/ min for 4 min, followed by a switch to 10% $O_2$ for 4 min. Gases were mixed with a static mixing element connected to mass flow controllers (Vögtlin Instruments). After 1-hr starvation on a food-free NGM plate, single adult (1 day post L4 larval stage) worms expressing both mCherry and GCaMP5K in AVK were loaded into the worm channel: Animals were transferred into a drop of S-Basal on the NGM plate. By applying a short vacuum to the worm outlet, we sucked animals up into Tygon tubing, which was afterwards connected again to the worm inlet to position the worm in the channel. We used an epifluorescence microscope equipped with a CoolLED pE-2 excitation system to provide dual wavelength excitation light using an ET-EGFP/mCherry filter set (59022x, Chroma) and dichroic (59022 bs, Chroma), and an Optosplit II (Cairns) image splitter (filter set used: 580 nm beam splitter and 520/35 nm and 641/ 75 nm bandpass emission filters). Split imaging data were acquired with an Andor iXon 397 EMCCD camera with 100 ms exposure time, streaming images at 10 Hz acquisition rate to MetaMorph software (Universal Imaging). Fluorescence values were measured with a custom-made tracking script written in MetaMorph software. The region of bright mCherry signal is detected by thresholding and robustly tracked using the built-in track object function. Measurement regions for GCaMP signal as well as nearby regions for measuring background values were manually selected for the first image frame. Their positions were updated according to the frame-to-frame displacement of the mCherry-tracking region. Normalized ratio dR/R (%) was calculated in the same way as for the freely moving calcium-imaging data.

## Worm culture and strains

Worms were maintained at 20°C on plates of agar nematode growth medium (NGM) seeded with OP50 *Escherichia coli* bacteria as a food source. Wild-type was *C. elegans* Bristol strain N2. Mutant strains used in this study were:

ZIM144, *flp-1(ok2811)IV*, 6x outcrossed to N2
ZIM550, *nlp-12(ok335)I*, 6x outcrossed to N2
ZIM551, *flp-1(ok2811)IV;nlp-12(ok335)I*, derived from crossing ZIM144 with ZIM550.

We verified the *flp-1(ok2811)* allele by cDNA sequencing: it deletes the last two bases of exon 1 and the entire exon 2 of the *flp-1* coding region resulting in an artificial exon 1 (made up of remaining unspliced parts of intron 2) with a predicted premature stop codon; further this causes a frame shift starting from exon 3 including the whole sequence section encoding the actual neuropeptides. Therefore *flp-1(ok2811)* is a prospective null allele. As opposed to the previously described alleles *flp-1(yn2)* and *flp-1(yn4)*, *flp-1(ok2811)* does not affect the nearby *daf-10* coding sequence.

Mutant worm strains were received from Caenorhabditis Genetics Center (CGC), Liliane Schoofs Laboratory and Chris Li Laboratory.

Transgenic animals were generated by injecting plasmid mixes into gonads of young adult hermaphrodites and generating heritable extra-chromosomal arrays. All injection mixes were prepared to yield 100ng/ul by adding empty pSM plasmids when necessary. When indicated, *Punc-122::gfp*, *Punc-122::dsRed* (expressed in so-called coelomocytes) or *Pmyo-3::mCherry* (expressed in body wall muscles) were used as co-injection markers.

| Strain | Genotype | Description |
|---|---|---|
| ZIM466 | *lite-1(xu-7)X; mzmEx300 [Pflp-1(AVK)::GCaMP5K; Pflp-1(AVK)::mCherry]* | AVK imaging line |
| ZIM626 | *lite-1(xu-7)X; mzmEx407 [Pflp-1(AVK)::GFP; Pflp-1(AVK)::mCherry]* | AVK gfp control imaging line |
| ZIM563 | *lite-1(xu-7)X; mzmEx365 [Pnlp12::GCaMP5K; Pnlp-12::wCherry]* | DVA imaging line |
| TRL144 | *lite-1(xu-7)X; [Pnlp12::GFP; Pnlp-12::wCherry]* | DVA gfp control imaging line |
| ZIM319 | *flp-1(ok2811)IV; mzmEx142 [Pflp-1(AVK)::flp-1::SL2::gfp; Pmyo-3::mCherry]* | *flp-1* rescue in AVK |
| ZIM837 | *nlp-12(ok335)I; mzmEx505 [Pnlp12::nlp12::SL2::gfp; Pflp-17::gfp]* | *nlp-12* rescue under endogenous promoter |
| ZIM1255 | *flp-1(ok2811)IV;nlp-12(ok335)I; mzmEx505 [Pnlp12::nlp12::SL2::gfp; Pflp-17::gfp]; mzmEx142 [Pflp-1(AVK)::flp-1::SL2::gfp; Pmyo-3::mCherry]* | *flp-1* rescue in AVK & *nlp-12* rescue under endogenous promoter |
| ZIM367 | *mzmEx249 [Pflp-1(AVK)::egl-1::SL2::gfp; Pflp-1::mCherry; Punc-122::dsRed]* | AVK- |
| ZIM625 | *mzmEx406 [Pnlp-12::p12::SL2::mCherry; Pnlp-12::p17::SL2::mCherry; Pnlp-12::mCherry; Punc-122::gfp]* | DVA- |
| ZIM627 | *mzmEx249 [Pflp-1(AVK)::egl1::SL2::gfp; Pflp-1::mCherry; Punc-122::dsRed]; mzmEx406 [Pnlp-12::p12::SL2::mCherry; Pnlp-12::p17::SL2::mCherry; Pnlp-12::mCherry; Punc-122::gfp]* | AVK-; DVA- |
| CX11697 | *kyIs536 [Pflp-17::p17::SL2::gfp elt-2::gfp]; kyIs538 [Pglb-5::p12::SL2::gfp; elt-2::mCherry]* | BAG- (also ASG-, Roger Pocock, pers. comm.) |

Pro-apoptotic genes used for cellular ablation were *egl-1* (*Conradt and Horvitz, 1998*), or *p12* and *p17*, which encode domains from split caspase 3 (*ced-3*) (*Chelur and Chalfie, 2007*).

GCaMP5k in mzmEx365 is codon-optimized for *C. elegans* (GenScript).

wCherry is codon-optimized mCherry and was kindly donated by Mei Zhen laboratory.

## Molecular biology and promoters for tissue specific expression

PCR-amplified DNA fragments of interest flanked by restriction sites were cloned into pSM vectors. Promoters were inserted via FseI and AscI sites while coding regions were usually inserted via NheI and Acc651 sites.

*Pflp-1(AVK)*: a 505bp fragment of *flp-1* promoter which extends from position −513 to −9 relative to the *flp-1* start codon, expressed in AVK only (*Altun-Gultekin et al., 2001*; *Nelson et al., 1998*).

*Pnlp-12*: 383bp fragment, directly upstream of the ATG start codon of the *nlp-12* gene, reported to be solely expressed in DVA (*Hu et al., 2011*) and kindly provided by the Josh Kaplan laboratory.

*flp-1* genomic region including the whole coding regions: 1414bp fragment beginning at start codon and including 127bp of 3'UTR.

*nlp-12* genomic region including the whole coding regions: 437bp fragment from start to stop codon.

## Acknowledgements

We thank Ev Yemini for providing and modifying MatLab code, the *Caenorhabditis* Genetics Center (CGC), Liliane Schoofs and Chris Li Laboratories for worm strains, the Mei Zhen and Josh Kaplan laboratories for plasmids, Shawn Lockery for advice and Martin Colombini for manufacturing of mechanical components. The research leading to these results has received funding from the European Community's Seventh Framework Programme (FP7/2007-2013)/ERC grant agreement number 281869 (acronym: *elegansNeurocircuits*) to MZ and the Research Institute of Molecular Pathology (IMP). The IMP is funded by Boehringer Ingelheim.

## Additional information

### Funding

| Funder | Grant reference number | Author |
|---|---|---|
| European Research Council | 281869 | Manuel Zimmer |

The funders had no role in study design, data collection and interpretation, or the decision to submit the work for publication.

### Author contributions

IH, JR, Conception and design, Acquisition of data, Analysis and interpretation of data, Drafting or revising the article; FM, Conception and design, Acquisition of data, Analysis and interpretation of data; SK, Developed MatLab code, Conception and design, Analysis and interpretation of data, Drafting or revising the article, Contributed unpublished essential data or reagents; HSK, Conception and design, Analysis and interpretation of data; RL, Acquisition of data, Analysis and interpretation of data; MS, Developed MatLab code, Contributed unpublished essential data or reagents; LT, Conception and design, Acquisition of data; MZ, Conception and design, Analysis and interpretation of data, Drafting or revising the article

### Author ORCIDs

Manuel Zimmer, http://orcid.org/0000-0002-8072-787X

## Additional files

### Supplementary files

• Source code 1. The zip archive contains source code written with MatLab to calculate skeletons and skeleton angles from binary worm images, perform eigenworm decomposition into undulation mode and turning mode, reconstruct worm images and movies from undulation mode and turning mode, and extract behavioral data with respect to spatial orientation (bearing direction, curving bias, reversals, speed, mean body turning). See 'readme.txt' for an overview.

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
