## [Decision Letter]

Thank you for submitting your work entitled "Regulation of two motor patterns enables the gradual adjustment of locomotion strategy in *Caenorhabditis elegans*" for consideration by *eLife*. Your article has been favorably evaluated by Eve Marder (Senior Editor) and three reviewers, one of whom, Ronald L Calabrese, is a member of our Board of Reviewing Editors.

The reviewers have discussed the reviews with one another and the Reviewing Editor has drafted this decision to help you prepare a revised submission.

Summary:

Hums et al. present a rigorous analysis of how *C. elegans* changes movement when oxygen drops concentration using Stephens' method. Movement is divided into "crawling" when posture fits the first two PCAs and "steering" for all other behavior. Worms steer more once oxygen drops, but this adapts in 3-minutes. Confirming previous reports, they show that two peptidergic-interneurons influence posture, and then show that disrupting the neurons or neuropeptide genes predictably effect posture with oxygen drop. The truly novel results image neuronal calcium in freely moving worms during oxygen drops. AVK activity roughly follows overall speed. DVA activity correlated with crawling and/or reversing, modes when the body is curved. Lastly, they find evidence for both pirouette and weathervane mechanisms for aerotaxis. The experiments are carefully performed and the data presented are clear and convincing. Writing is succinct and clear. Supplemental data are extensive but could be better integrated into the manuscript. Methods are described in detail.

This work is an inventive combination of computational and biological methods that adds to our understanding of the subtleties of *C. elegans* navigation, at both the behavioral and mechanism level. It provides a clear quantitative description for how the worm adapts in neural function and behavior to new oxygen conditions using two neuropeptides with opposing function.

Essential revisions:

1) The authors should rethink their presentation and be more circumspect about linking this work to ARS work. We are concerned that locomotory transitions as studied in this paper are different enough in conditions and timing from the ARS as described in Hills et al. (2004) that the behaviors might be different: i.e., use different neurons and genes. The Zimmer study does not test the same genetic or neuronal manipulations in the Hills paper nor in previous roam/dwell papers. So the neuronal and genetic mechanisms may or may not share mechanisms completely or even substantively. This hasn't been explicitly tested. The switch to low oxygen may simply impede movement transiently before some physiological compensation mechanism kicks in allowing the worms to crawl well again. If this is the case, then the behavior the authors are studying has little to do with "search" per se. We ask that the authors leave room for readers to understand that it remains to be tested whether the genes and neurons are the same or different between the similar looking behaviors.

2) The method of Stephens is a powerful one in this analysis that allows separation of different aspects of locomotion. The authors should however be more precise in their terminology and not refer to the separated Eigen-worms as representing crawling vs. steering. These are more precisely termed a transnational component vs. a directional component of locomotion rather than distinct behaviors.

3) The paper will be improved if the data on rescue experiments are moved to the main figures. For simplicity sake, there might be way to do this by modifying existing main figures, but adding new figures is also an option. Remember that *eLife* publishes supplements to figures as well, so please integrate all figures into the text in one or another fashion.

4) The authors have already succeeded in rescuing single mutants. So they have the reagents in hand for double mutant rescues. If they could rescue at least one (or two) genes in the double mutant that would provide evidence that the lack of response in the double mutant is due to mutations in these two genes and not due to background mutations in the double mutant.

---

## [Author Response]

*1) The authors should rethink their presentation and be more circumspect about linking this work to ARS work. We are concerned that locomotory transitions as studied in this paper are different enough in conditions and timing from the ARS as described in Hills et al. (2004) that the behaviors might be different: i.e., use different neurons and genes. The Zimmer study does not test the same genetic or neuronal manipulations in the Hills paper nor in previous roam/dwell papers. So the neuronal and genetic mechanisms may or may not share mechanisms completely or even substantively. This hasn't been explicitly tested. The switch to low oxygen may simply impede movement transiently before some physiological compensation mechanism kicks in allowing the worms to crawl well again. If this is the case, then the behavior the authors are studying has little to do with "search" per se. We ask that the authors leave room for readers to understand that it remains to be tested whether the genes and neurons are the same or different between the similar looking behaviors.* We fully appreciate this concern about linking previous work on ARS after removal from food with oxygen-induced ARS presented in our study. The conditions in the two paradigms are indeed very different. We resolved this confusion by clearly distinguishing the two paradigms as ‘ARS after removal from food’ (previous studies) and‘O_2_-induced ARS’ (our study) throughout the revised text.

We still like to use the term ARS due to the behavioral features expressed after the O_2_ stimulus, which are largely overlapping with the behavioral features after removal from food. However, we included a paragraph in the Discussion (second paragraph) making clear that, besides NLP-12 neuropeptides and DVA neurons expressing them (study from Bhattacharya et al., 2014), no other neurons or genes from studies on ARS after removal from food have been analyzed here in the O_2_-downshift paradigm; therefore we do not conclude that the two behaviors are controlled by the same interneuron circuits with the exception of DVA. Furthermore, we include a small section in the Discussion (fourth paragraph) explaining that O_2_-induced ARS of food-deprived animals and roaming-dwelling decisions in the presence of food likely employ different modulatory signaling pathways.

*2) The method of Stephens is a powerful one in this analysis that allows separation of different aspects of locomotion. The authors should however be more precise in their terminology and not refer to the separated Eigen-worms as representing crawling vs. steering. These are more precisely termed a transnational component vs. a directional component of locomotion rather than distinct behaviors.* We agree that the term ‘crawling’ is already taken to describe general worm movement on agar. We decided instead to use the term ‘undulation mode’ for the motions captured by the first two eigenworms (EWs) as the combination of EW1 and 2 corresponds to the regular undulatory movements along the worm’s body, which generates forward and backward crawling.

Likewise, we agree that the term ‘steering’ implies orientation towards a goal and is not appropriate to describe behavior in response to temporal sensory stimuli, but in the absence of a spatial gradient. We therefore changed to the term ‘turning mode’.

*3) The paper will be improved if the data on rescue experiments are moved to the main figures. For simplicity sake, there might be way to do this by modifying existing main figures, but adding new figures is also an option. Remember that* eLife

*publishes supplements to figures as well, so please integrate all figures into the text in one or another fashion.*

This is a very good suggestion to highlight the importance of the neuropeptide signaling pathways of this study. We therefore converted the figure with rescue experiments to a main figure (Figure 6).

4) The authors have already succeeded in rescuing single mutants. So they have the reagents in hand for double mutant rescues. If they could rescue at least one (or two) genes in the double mutant that would provide evidence that the lack of response in the double mutant is due to mutations in these two genes and not due to background mutations in the double mutant.

To analyze whether the compromised modulation of undulation and turning mode in *flp-1;nlp-12* double mutant animals were indeed caused by *flp-1* and *nlp-12* mutant alleles, but not by a combination of background mutations, we generated a new worm strain by intercrossing the single mutant rescue strains. In the revised manuscript we include results on rescuing simultaneously *flp-1* in AVK and *nlp-12* in DVA in *flp-1;nlp-12* double mutant animals. In this double rescue strain, similarly as in the single rescue strains, the behavioral features are partially restored. See Figure 6 and Figure 5—figure supplement 1in the revised manuscript.